# Loss of SUMO-specific protease 2 causes isolated glucocorticoid deficiency by blocking adrenal cortex zonal transdifferentiation in mice

Damien Dufour [1], Typhanie Dumontet[1,2,3], Isabelle Sahut-Barnola[1], Aude Carusi[4], Méline Onzon[1], Eric Pussard[5], James Jr Wilmouth[1], Julie Olabe [1], Cécily Lucas[1,6], Adrien Levasseur[1], Christelle Damon-Soubeyrand[1], Jean-Christophe Pointud[1], Florence Roucher-Boulez [1,6], Igor Tauveron[1,7], Guillaume Bossis [4], Edward T. Yeh[8], David T. Breault [9,10], Pierre Val[1], Anne-Marie Lefrançois-Martinez[1] & Antoine Martinez [1] ✉

SUMOylation is a dynamic posttranslational modification, that provides fine-tuning of protein function involved in the cellular response to stress, differentiation, and tissue development. In the adrenal cortex, an emblematic endocrine organ that mediates adaptation to physiological demands, the SUMOylation gradient is inversely correlated with the gradient of cellular differentiation raising important questions about its role in functional zonation and the response to stress. Considering that SUMO-specific protease 2 (SENP2), a deSUMOylating enzyme, is upregulated by Adrenocorticotropic Hormone (ACTH)/cAMP-dependent Protein Kinase (PKA) signalling within the *zona fasciculata*, we generated mice with adrenal-specific *Senp2* loss to address these questions. Disruption of SENP2 activity in steroidogenic cells leads to specific hypoplasia of the *zona fasciculata*, a blunted reponse to ACTH and isolated glucocorticoid deficiency. Mechanistically, overSUMOylation resulting from SENP2 loss shifts the balance between ACTH/PKA and WNT/β-catenin signalling leading to repression of PKA activity and ectopic activation of β-catenin. At the cellular level, this blocks transdifferentiation of β-catenin-positive *zona glomerulosa* cells into *fasciculata* cells and sensitises them to premature apoptosis. Our findings indicate that the SUMO pathway is critical for adrenal homeostasis and stress responsiveness.

The murine adrenal cortex is a constantly self-renewed endocrine organ composed of concentric zones, including the outermost *zona glomerulosa* (zG) layer producing mineralocorticoids and the innermost *zona fasciculata* (zF) layer producing glucocorticoids. According to the centripetal migration model occurring during postnatal development, progenitors cell populations located in the adrenal capsule (characterised by GLI1 expression) or within the zG (characterised by

SHH expression) consecutively differentiate into steroid-producing zG cells, then through a process of zonal transdifferentiation, convert into zF cells before eventually undergoing apoptosis at the corticomedullary junction[1–3].

Genetic models in mouse and in vitro approaches have identified two important signalling pathways for adrenal cortex homoeostasis. On the one hand, the WNT/Rspondin/β-catenin pathway is necessary

for the maintenance of progenitor pools and the acquisition of zG identity[4–6]. On the other hand, cAMP/PKA signalling, stimulated by pituitary ACTH, triggers the recruitment of progenitors by inducing the transdifferentiation of zG cells into zF cells and stimulates glucocorticoid production[7,8]. We and others have previously shown that these two signalling pathways antagonise each other by modulating various actors such as WNT4, PDE2A or CCDC80[3,6,9]. Optimal response to PKA signalling and therefore zF differentiation is also subject to epigenetic programming by the histone methyl transferase EZH2[10]. Nonetheless, the mechanisms that maintain adrenal cortex zonation and balance between these two pathways are yet to be discovered.

SUMOylation is a dynamic and one of the fastest evolving[11] post-translational modification consisting in the covalent addition of SUMO peptides on a target protein. This modification can affect various processes such as protein stability, interactions or subcellular localisation[12]. There are three main SUMO peptides in rodents, namely SUMO1, which shares around 50% of identity with SUMO2 and SUMO3, the two latter differing by only three amino acids. SUMOylation is achieved through an enzymatic cascade involving activation by the E1 heterodimer (SAE1/UBA2), conjugation by the sole E2 enzyme UBC9 (encoded by *Ube2i*) and final ligation by various E3 SUMO ligases such as members of the PIAS and TRIM families as well as RANBP2 and CBX4[13,14]. SUMO peptides can be removed from SUMO conjugated substrates by deSUMOylases belonging mainly to the Sentrin-specific proteases family (SENPs) or by the more recently discovered DeSI-1 and USPL1 making this posttranslational modification highly dynamic (Supplementary Fig. S1a, b). Several in vitro and in vivo studies have highlighted the importance of controlling SUMOylation levels to enable differentiation or maintain cellular identity[15–17] and tissue homoeostasis in vivo in various cell lineages[18,19]. The adrenal gland could provide a paradigm to study how SUMOylation dynamics can interact with the function and homoeostasis of an organ, in charge of constant adaptation to stress.

We have previously shown that protein SUMOylation follows a decreasing centripetal gradient in human and mouse normal adrenal cortices. Moreover, this gradient is altered in genetic endocrine diseases with deregulated PKA or WNT signalling pathways[20]. Remarkably, SUMOylation is negatively and acutely regulated by ACTH in both adrenal cortex and adrenocortical cell cultures through transcriptional control of key enzymes, especially SENP2 whose upregulation by PKA correlates with transient hypoSUMOylation in zF. Interestingly, PKA-mediated upregulation of *Senp2* was previously shown to promote the progression of preadipocytes into the adipogenic programme[15]. Taken together, these studies suggest that limiting SUMOylation may facilitate or be a prerequisite for any change in differentiation states. Conversely, an excess of WNT/β-catenin signalling in the adrenal cortex induces an expansion of zG identity that is correlated with a high SUMOylation state[20]. Finally, preventing in vivo SUMOylation of the transcription factor SF-1 (*SF-1^{2KR/2KR}* mice), the main driver of adreno-gonadal cell fate, disturbs endocrine development by maintaining discrete gonadal traits in the cortex and adrenal traits in the testis[21]. This highlights the need to control SUMOylation during cell fate decisions leading to adrenal cortex identity.

We hypothesise that disruption of the SUMOylation gradient in the adrenal cortex may disrupt zonation and impair adaptive response to stress. In order to understand the implication of the SUMO pathway on homoeostatic maintenance and endocrine function, we have developed mouse models of adrenal hyperSUMOylation by conditional ablation of *Senp2* in the cortex (*Senp2^{cKO}*). Our report reveals that *Senp2^{cKO}* mice show zone-specific adrenal atrophy, isolated glucocorticoid deficiency and blunted response to ACTH. Progressive atrophy of zF evoked by SENP2 deficiency results from a blockade of zonal transdifferentiation, early apoptosis and impaired PKA catalytic activity that cannot be rescued by genetic derepression of the PKA holoenzyme. SENP2-deficient adrenals also show increased β-catenin

SUMOylation and activity that may help to antagonise PKA signalling, thus maintaining the suppression of zF identity. As *Senp2* expression is itself under the control of ACTH/PKA, our data identify SUMOylation as a feedforward mechanism that readies the adrenal cortex to respond to stress and maintain functional zonation.

## Results

### Senp2 invalidation in the adrenal cortex leads to zF hypoplasia and adrenal dysplasia

To assess the role of SUMOylation in the adrenal cortex, we have developed a mouse model with specific deletion of the ACTH-regulated deSUMOylase SENP2[20] in steroidogenic cells by mating *Senp2^{fl/fl}*[22] mice with *Sf-1(Nr5a1)*-Cre mice[23]. *Senp2* conditional knockout mice are later referred to as *Senp2^{cKO}*. *Senp2* deletion was confirmed in 4-week-old mouse adrenals by RT-qPCR analyses showing reduced *Senp2* mRNA accumulation in both genders. Western blot analysis confirmed reduced SENP2 protein accumulation and genomic PCR, confirmed adrenal-specific recombination at the *Senp2* locus (Supplementary Fig. S1c–e). Monitoring of adrenal mass from 4 to 40 weeks of age revealed significant adrenal hypoplasia in mutants, occurring between 4 and 8 weeks in both sexes. After this time point, overall adrenal weight in *Senp2^{cKO}* remained below that of controls in females only because of sex differences in the kinetics of adrenal mass gain (almost continuous growth in WT females contrasting with a progressive decrease in WT males over time) (Fig. 1a). To investigate the causes of this hypoplasia, we performed H&E staining, which revealed two different histological phenotypes: either homogeneous atrophy of the cortex leaving medulla centrally located, or cortical atrophy accompanied by dysplasia due to clusters of large eosinophilic cells, usually at one pole of the gland and pushing the medulla toward the other pole (Fig. 1b, left panel).

Immunofluorescence co-staining for zonal markers showed that, compared to WT, the integrity of zG (DAB2 +) did not seem affected in *Senp2^{cKO}* adrenals of 8-week-old mice, while zF (AKR1B7 +) was atrophic and sometimes mislocated together with the medulla (TH +) in dysplastic glands (Fig. 1b, right panels). To confirm that the zF was the most affected by *Senp2* invalidation, the number of cells in each cortical zones and medulla was counted on 2D sections. This showed a dramatic and specific reduction in total cortical cell number regardless of sex, solely attributable to the 75–80% loss of zF cells (Fig. 1c and Supplementary Fig. S1f). Laminin immunostaining was then used to examine vascular architecture in WT and mutant adrenal sections. Typical capillaries surrounding the "rosette" structures in the zG and delimiting the zF cells columns were evidenced in WT adrenal sections. Consistent with zonal markers immunostaining, a rosette-like pattern of vascularisation, typical of zG, was found in areas with atrophic zF (Fig. 1b*a*), whereas large columnar structures surrounding clusters of hypertrophic zF cells were evidenced in dysplastic *Senp2*-deficient glands (Fig. 1b*b*).

To further characterise the origin of adrenal dysplasia, we introduced the *Rosa26RmTmG* reporter transgene[24] (Supplementary Fig. S1g) into *Senp2^{cKO}* background to trace *Senp2* recombination using GFP immunostaining and non-recombined cells using Tomato immunostaining. WT adrenals (from *Sf1-Cre/+::Senp2^{fl/+}::R26R^{mTmG/mTmG}* triple transgenic mice) presented full recombined cortex, while in dysplastic *Senp2^{cKO}* adrenals (*Sf1-Cre/+::Senp2^{fl/fl}::R26R^{mTmG/mTmG}*), most of the cells belonging to the mislocated zF were GFP-negative/Tomato-positive, hence non-recombined (Fig. 1d). To address the identity of these cells escaping recombination, we performed co-staining of GFP with the canonical steroidogenic marker SF-1. Surprisingly, GFP-negative cells expressed SF-1, implying that they kept full steroidogenic identity (Fig. 1d, bottom centre panel) as also suggested by detection of zonal markers (Fig. 1b, right panels, bottom). To confirm that these cells were hypertrophic as H&E and Laminin staining suggested, we measured 2D cell areas in the zF of WT and *Senp2^{cKO}* adrenal sections. This showed

that in females, although GFP-positive and GFP-negative zF cell areas did not differ in the mutant, GFP-negative cells were larger than zF cells from WT adrenals (Fig. 1e). Similar observations were made in males but with a *P* value of 0.0560. Next, we assessed global SUMOylation

status by western blot and found no variation in the profile of SUMO1 or SUMO2/3 conjugates as a function of the genotype (Supplementary Fig. S1h, i). However, GFP-positive zF cells retained nuclear SUMO2/3 staining whereas GFP-negative had diffuse staining in the cytoplasm

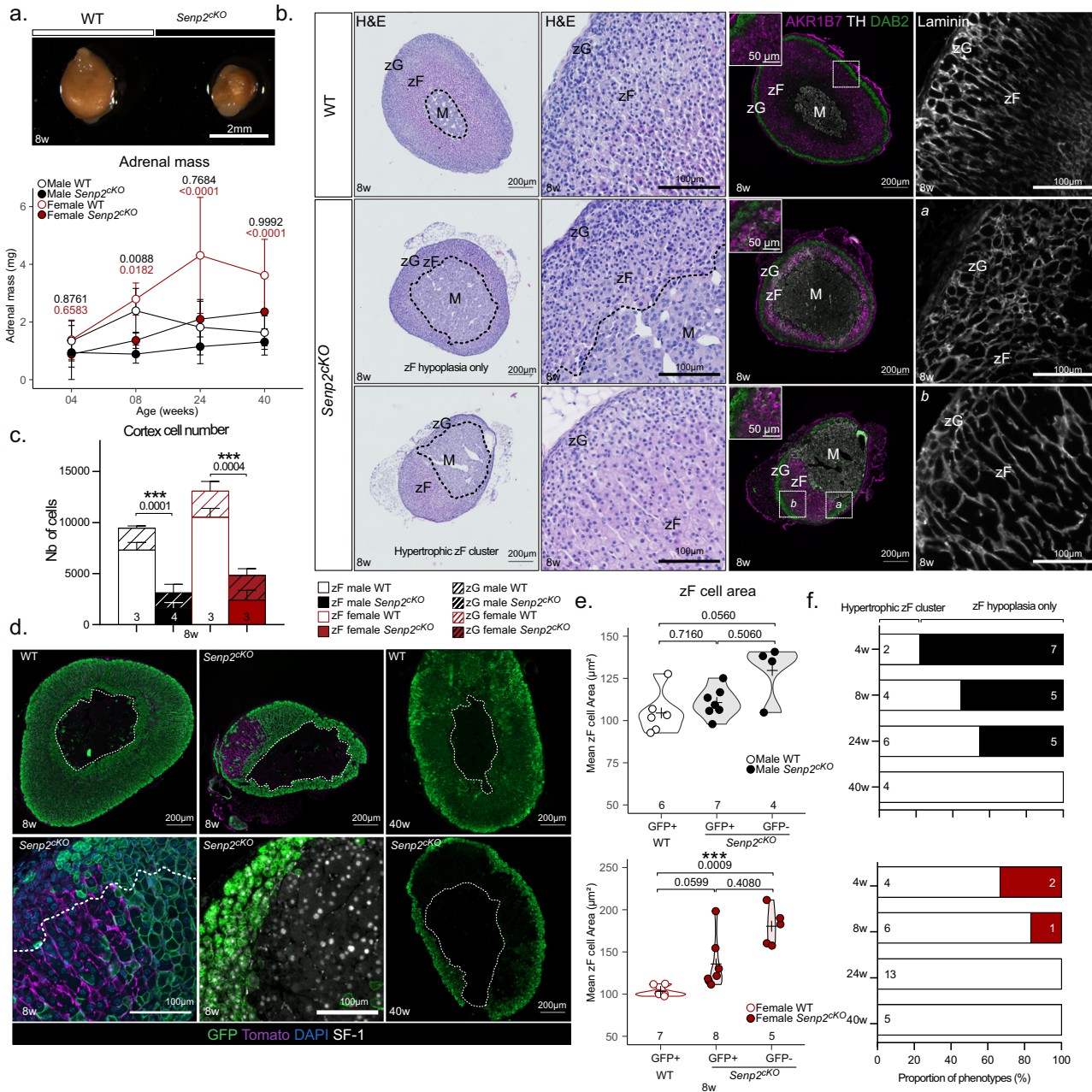

**Fig. 1 | Deletion of the deSUMOylase *Senp2* in embryonic steroidogenic cells induces adrenal dysplasia and zF hypoplasia in adult mice. a** Top: Representative picture of an 8-week-old WT (left) and a *Senp2^cKO* (right) adrenal. Bottom: Absolute adrenal mass follow-up of male (*n* = 45, 23, 15 and 32 adrenals for WT and 37, 9, 14 and 15 for *Senp2^cKO*) and female (*n* = 40, 28, 13 and 44 adrenals for WT and 34, 7, 7 and 27 for *Senp2^cKO*) mice WT and *Senp2^cKO* (mean) from 4 to 40 weeks of age. *P* values represent the difference between genotypes within the same age and sex group. Data are presented as mean values +/− SD. **b** Morphological analysis of the phenotypes on 8-week-old males' adrenals. Left: H&E staining of WT or *Senp2^cKO* adrenals with or without dysplasia. Centre: Coimmunofluorescence labelling of AKR1B7 (purple), TH/Tyrosine Hydroxylase (white) and DAB2/Disabled2 (green) in WT and *Senp2^cKO* adrenals. Right: Immunofluorescence analysis of laminin (white) revealing the vascular network in WT and cKO adrenals. **c** 2D cell number counting in male and female adrenals at 8 weeks of age. (*n* = 4 adrenals for *Senp2^cKO* males and

*n* = 3 for other conditions). Data are presented as mean values +/− SEM. *P* values were determined by two-sided Mann−Whitney test. **d** Coimmunofluorescence labelling of GFP (green) with Tomato (purple) and SF-1 (white) at 8 (left and centre) and 40 weeks of age (right) WT and *Senp2^cKO* adrenals. **e** Morphometric analysis of zF cells area in WT (GFP+) adrenals and recombined (GFP+) or non-recombined (GFP-) zones of *Senp2^cKO* adrenals at 8 weeks of age. *P* values were obtained with Kruskal−Wallis test and adjusted with FDR method. **f** Prevalence of phenotypes in cKO adrenals at different ages. Black (male) or red (female) bars represent hypoplastic adrenals without any gross morphological change and white bar represents hypoplastic and dysplastic adrenals harbouring hypertrophic zF clusters. zG *zona glomerulosa*, zF *zona fasciculata*, WT wild-type, cKO conditional knockout. *P value < 0.05; **P value < 0.01; ***P value < 0.001. Source data are provided as a Source Data file.

(Supplementary Fig. S1j, k). Together, these results indicate that the clusters of cellular hypertrophy found in dysplastic *Senp2^cKO* adrenals are predominantly composed of zF cells that have escaped *Senp2* ablation and that this phenotype (hypertrophic zF clusters) seems more likely to occur in females. To examine further this sexual dimorphism, we assessed the proportion of mutant adrenals presenting with hypertrophic zF clusters from 4 to 40 weeks of age in both genders (Fig. 1f). Throughout time, the frequency of adrenals with hypertrophic zF clusters increased in a sexually dimorphic fashion: indeed, for these clusters to develop in all adrenals, it took 40 weeks in males and only 24 in females. Thus, in 40-week-old mice, *Senp2^cKO* adrenal cortex contained, compared to WT, very few GFP-positive zF cells, with GFP staining limited to zG and non-recombined cells constituting the entire zF (Fig. 1d, right panel and Supplementary Fig. S1l).

SF-1 is not expressed in corticotrope cells of the pituitary (Supplementary Fig. S2a) but is expressed in gonadotropes (Supplementary Fig. S2b)[25] and somatic cells of the gonads[23]. In consequence, we measured plasmatic gonadotropins and sex steroids levels. Compared to WT, 8-week-old *Senp2^cKO* males had lower levels of follicle-stimulating hormone (FSH) but unchanged levels of luteinising hormone (LH) and testosterone (Supplementary Fig. S2c–e). Mutant females, on the other hand, had higher levels of LH whereas FSH and progesterone levels were unaltered (Supplementary Fig. S2g–i). The mass of testis was reduced in *Senp2^cKO* mice and some tubules showed disorganisation of seminiferous epithelium (Supplementary Fig. S2f, k). Mutant ovaries were smaller than the WT and did not contain any *corpora lutea* (Supplementary Fig. S2j, l) which could explain why mutant female were hypofertile with a breeding success rate of less than 25% versus 75% for WT females (Supplementary Fig. S2m). These results show that SENP2 has a deleterious effect on reproductive function that will be investigated in more detail in future studies. Thus, gonadal phenotype is unlikely to explain adrenal cortex atrophy in *Senp2^cKO* mice, although we did not exclude an indirect effect of sex steroids to explain the better ability of female adrenals to escape *Senp2* ablation.

In conclusion, these results show that SENP2 is necessary for proper adrenal cortex zonation and homoeostatic maintenance. Indeed, its inactivation initially leads, to zF atrophy, which is compensated over time by the recruitment of cells escaping *Senp2* recombination allowing them to maintain a wild-type *Senp2* zF in otherwise mutant adrenals. Importantly, we show that females more efficiently overcome zF atrophy induced by *Senp2* loss. This further highlights the sexually dimorphic traits of adrenal homoeostasis[26,27].

## Loss of SENP2 is associated with isolated glucocorticoid deficiency

Given the profound alterations and notably, the time and sex-dependant remodelling of adrenal cortex zonation in *Senp2^cKO*, we measured changes in circulating steroid levels and assessed steroidogenic gene expression. Steroid hormones are synthesised from cholesterol through enzymatic processes resulting in corticosterone production by zF cells and aldosterone by zG cells (Fig. 2a). Since zF was the most impacted zone by the *Senp2* mutation, we anticipated a reduction in plasmatic corticosterone levels, the main glucocorticoid in rodents. Indeed, compared to control males, corticosterone levels were dramatically reduced in *Senp2^cKO* at 4 weeks and remained lower at 8 weeks of age. Interestingly, levels normalised over time to be indistinguishable from controls by 24 weeks (Fig. 2b, top). In contrast, *Senp2* inactivation had no impact on corticosterone concentrations in females, at any time point (Fig. 2b, bottom). The integrity of the zF and the corticosterone production are under the strict control of pituitary ACTH that maintains homoeostasis through a negative feedback loop mediated by the glucocorticoids on the hypothalamic-pituitary-adrenals (HPA) axis. Therefore, we measured circulating ACTH in

*Senp2^cKO* and found a eight- to tenfold increase at 24 weeks in both sexes. This is consistent with a dysfunction of the zF cells, resulting in a subclinical insufficiency over time (Fig. 2c).

For insights into the mechanisms of this insufficiency, we analysed the expression of steroidogenic genes in adrenals of 4-week-old mice. RT-qPCR analyses showed that mRNA levels of *Star* and *Cyp11a1* encoding rate-limiting step proteins in steroidogenesis, were decreased in both male and female *Senp2^cKO* adrenals (Fig. 2d). Interestingly, *Cyp21a1*, *Hsd3b1* and *Cyp11b1* transcripts were specifically downregulated in female *Senp2^cKO* adrenals (Fig. 2d). To assess the impact of *Senp2* loss on zG function, we measured mineralocorticoids plasma levels and *Cyp11b2* expression. Consistently with histological observations, we did not find any negative effect of *Senp2* ablation on aldosterone levels, but rather a positive trend with an increase in 18-hydroxy-corticosterone plasmatic concentration in *Senp2^cKO* males at 24 weeks and a trend toward upregulation of *Cyp11b2* expression in *Senp2^cKO* female adrenals at 4 weeks (Supplementary Fig. S3a–c).

To further evaluate the steroidogenic capacity of *Senp2*-deficient cells, we assessed the enrichment of neutral lipids (mainly consisting in cholesteryl esters, available precursor for steroid synthesis) in steroidogenic cells from 40-week-old adrenals. Bodipy staining revealed high lipid accumulation in zF cells from WT adrenals and GFP-negative hypertrophic cells from *Senp2^cKO* adrenals, whereas lipid accumulation was lower in nearby GFP-positive hypoplastic cells (Fig. 2e).

In conclusion, these results draw a picture of the differential role of *Senp2* in the zonation of the adrenal cortex. While it is dispensable for the zG, its absence leads to a deficient zF struggling to produce enough corticosterone to maintain homoeostasis. This effort is illustrated by elevated circulating ACTH and hypertrophic zF cells, hallmarks of isolated glucocorticoid deficiency.

## Adrenal cortex lacking *Senp2* shows blunted ACTH response

To understand the underpinnings of the adrenal phenotype in *Senp2^cKO* mice, knowing that ACTH is a regulator of SUMOylation in the adrenal cortex[20], we assessed the endocrine and transcriptional steroidogenic responses to acute ACTH stimulation. Plasma concentrations of steroids associated with glucocorticoid metabolism (Fig. 2a) were determined by LC–MS/MS, in 24-week-old mice injected 2 h before with ACTH and compared to vehicle (Fig. 3a and Table 1). We confirmed that, at this age, there were no differences in plasma levels of all steroids measured in basal conditions (vehicle) between WT and *Senp2^cKO* mice. As expected, ACTH treatment induced a strong increase in adrenal steroids (e.g., corticosterone levels were three- to sevenfold induced in females and males, respectively) in plasma from WT mice (except progesterone in females, which comes mainly from the ovaries). However, the ACTH-stimulated endocrine response was heavily blunted and at least halved in *Senp2^cKO* mice (Table 1). To determine whether alteration of endocrine response correlated with changes in gene expression, we measured mRNA levels of ACTH-responsive genes involved in the initial steps of steroidogenesis (i.e., *Scarb1* and *Star*) by RT-qPCR (Fig. 3b).

In males, *Senp2* mutation did not alter *Scarb1* and *Star* transcriptional responsiveness to ACTH (1.5- to twofold induction after 2 h) but impaired their basal expression, so that mRNA levels in ACTH-treated mutant adrenals barely reached basal expression in controls. By contrast in females, although their basal expression remained unchanged, both *Scarb1* and *Star* genes entirely failed to respond to ACTH in *Senp2^cKO* adrenals (Fig. 3b). To test whether this blunted transcriptional response relied on changes in phosphorylation of PKA substrates (Supplementary Fig. S4a), we performed western blots on trans-acting factor CREB and SUMO E3 ligase TRIM28, in response to 30 min ACTH treatment, in WT and mutant mice. Ser133 CREB and Ser473 TRIM28 phosphorylation levels were similar in basal conditions, increased in WT upon ACTH stimulation but failed to respond to treatment in *Senp2^cKO* adrenals (Fig. 3c). This impaired response to ACTH/PKA-

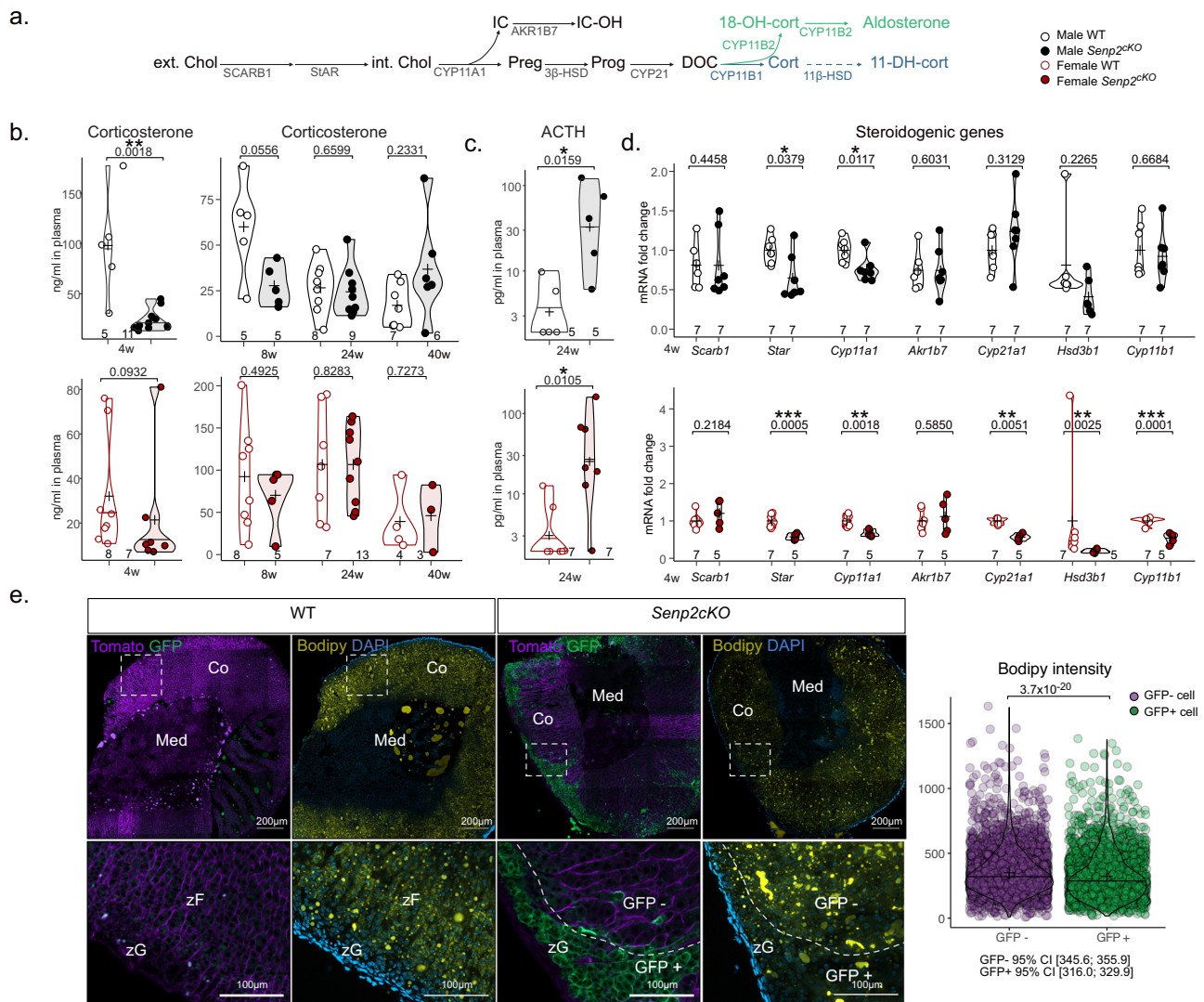

**Fig. 2 | Senp2 invalidation induces isolated glucocorticoid insufficiency.**
**a** Schematic representation of gluco- (blue) and mineralo- (green) corticoids synthesis: ext.Chol/int.Chol extra/intracellular cholesterol, Preg pregnenolone, Prog progesterone, DOC 11-deoxy-corticosterone, cort corticosterone, 18-OH-cort 18hydroxycorticosterone, including detoxication of cholesterol side-chain clivage IC (Isocaproaldehyde) into IC-OH (isocapryl alcohol) and inactivation of corticosterone into 11-DH-cort (11-dehydrocorticosterone). **b** Plasmatic concentration of corticosterone in WT and *Senp2^cKO* at 4 (determined by ELISA), 8, 24 and 40 weeks of age (determined by LC–MS/MS). *P* values were determined by two-sided *t* test for normally distributed condition or two-sided Mann–Whitney test. **c** ACTH plasmatic

levels of 24-week-old WT and *Senp2^cKO* mice. *P* values were determined by two-sided *t* test for normally distributed condition or two-sided Mann–Whitney test. **d** qPCR analyses of steroidogenic genes mRNA accumulation in 4-week-old WT and *Senp2^cKO* mice. *P* values were determined by two-sided *t* test for normally distributed condition or two-sided Mann–Whitney test. **e** Endogenous expression of GFP (green) and Tomato (purple) with Bodipy staining on WT and *Senp2^cKO* adrenal cortices at 40 weeks of age. *P* values was determined by two-sided Mann–Whitney test. *P* value < 0.05; **P* value < 0.01; ***P* value < 0.001. Source data are provided as a Source Data file.

mediated phosphorylation was unlikely caused by altered expression of ACTH receptor and co-receptor (*Mc2r* and *Mrap*, respectively) that were unaltered by *Senp2* loss (Supplementary Fig. S4b). To determine whether the deficiency in ACTH response involved PKA holoenzyme or occurred upstream of the kinase, we assessed the capacity of genetic activation of PKA to rescue adrenal insufficiency in *Senp2^cKO* mice by removing the RIα subunit (*Prkar1a* floxed allele) known to repress PKA catalytic activity[7,9,28]. As expected, 4-week-old *Prkar1a^cKO* mice developed large adrenals with hyperplastic zF and atrophic zG (Supplementary Fig. S4c, d). Adrenals from *Senp2,Prkar1a^dcKO* and *Senp2^cKO* mice were both dysplastic and showed reduction in cortical cell numbers. However, *dcKO* adrenals showed an atrophic zG (loss of DAB2 staining) presumably resulting from the antagonistic action of PKA signalling on zG identity (Fig. 3d and Supplementary Fig. S4c, d)[7,9]. As a result, in the absence of NaCl supplementation, *Senp2,Prkar1a^dcKO*

mice died prematurely from salt wasting (Supplementary Fig. S4e), whereas *Senp2^cKO* mice only suffered from isolated glucocorticoid deficiency. Thus, genetic derepression of PKA was unable to overcome cortical atrophy and dysplasia imparted by *Senp2* deficiency. This suggested that consequences of *Senp2* loss, including the excess of SUMOylation, had a dominant impact on the zF homoeostasis over PKA constitutive activation. Then, we explored endocrine activity of double KO mice. Although plasma corticosterone dosage showed no differences among the four genotypes, plasma ACTH concentrations were elevated in *Senp2^cKO* and reduced in *Prkar1a^cKO*, consistent with the corresponding associated disorders i.e. glucocorticoid deficiency and ACTH-independent glucocorticoid excess, respectively (Fig. 3e, f). By contrast, *Senp2-Prkar1a* double ablation restored ACTH levels to control values. This strongly suggested that the lack of *Senp2* resulted, among other things, in the repression of PKA catalytic activity that the

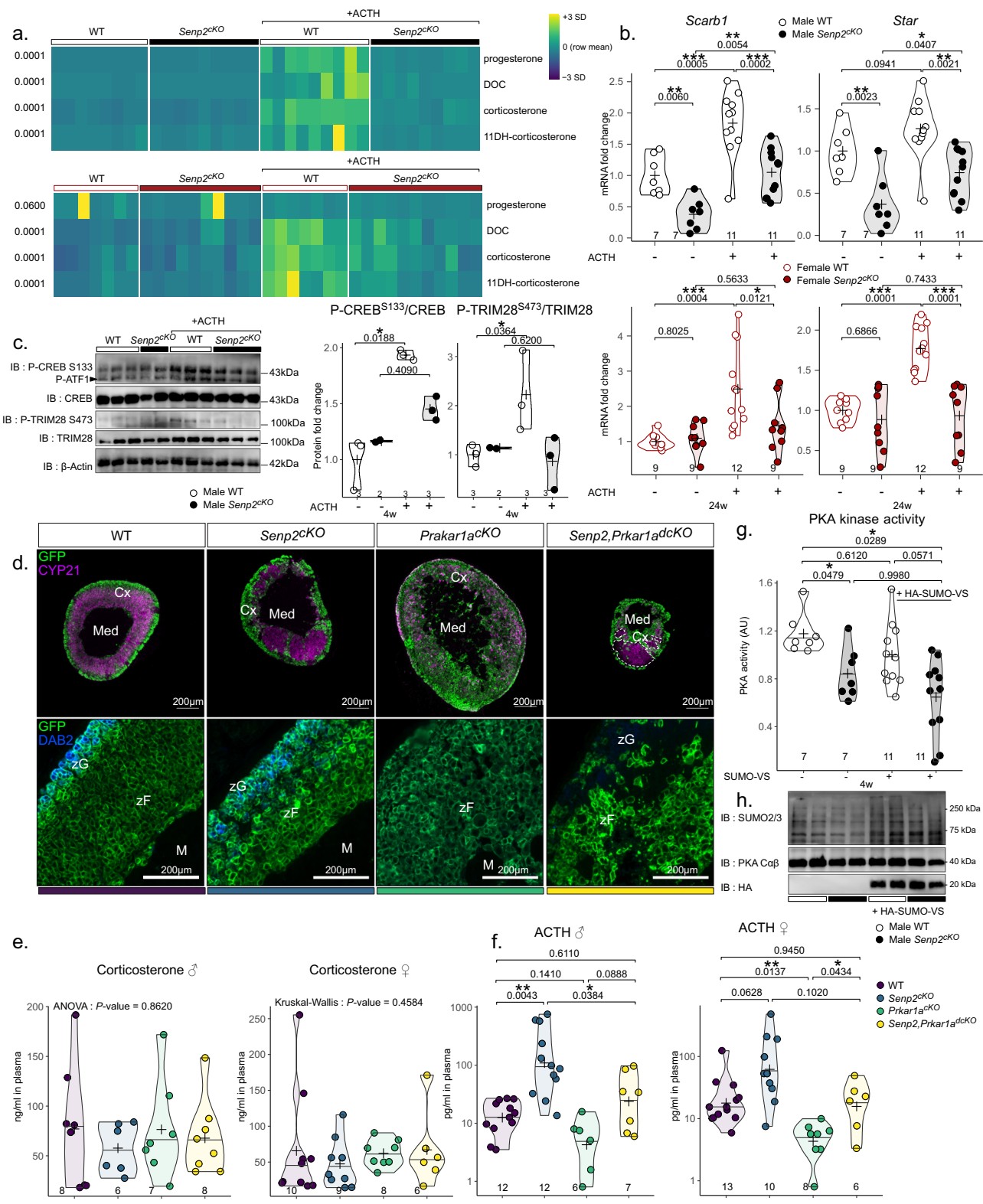

deletion of RIα regulatory subunits could partially overcome. Indeed, PKA kinase activity was decreased by 30% in *Senp2^cKO* adrenal extracts compared to WT without affecting Cαβ protein levels (Fig. 3g, h). To test a possible direct repressive effect of SUMOylation, we measured kinase activity in the presence of a mix of SUMO1/2 modified with vinyl sulfone (SUMOs-VS) acting as specific trap and potent inhibitors of SENPs SUMO proteases present in the extracts[29]. Under these conditions enhancing SUMOylation in adrenal extracts, PKA kinase activity

was further decreased in *Senp2^cKO*, reaching a 50% inhibition (Fig. 3g, h). Together, these results present SENP2 as a mandatory actor of proper ACTH response acting most likely by limiting repressive action of SUMOylation on the catalytic activity of the PKA holoenzyme.

### SENP2 loss represses PKA activity through hyperSUMOylation
To test the hypothesis that reduced PKA activity was at least caused by SUMOylation of catalytic subunits, we analysed primary sequences of

**Fig. 3 | Senp2 is necessary for proper ACTH response. a** Heatmap representing the plasmatic concentration of progesterone, DOC, corticosterone and 11-dehydrocorticosterone after treatment with PBS or ACTH for 2 h determined by LC-MS/MS in WT or *Senp2^cKO* 24-week-old mice. **b** qPCR analysis of ACTH-responsive genes mRNA accumulation in 24-week-old WT and *Senp2^cKO* mice after treatment with PBS or ACTH for 2 h. *P* values were obtained from one-way ANOVA and adjusted with FDR method. **c** Western blot analysis of phosphorylated CREB (Ser133) and TRIM28 (Ser473) in 4-week-old mice treated with PBS or ACTH for 30 min. Graphs represent phosphorylated form over total form. *P* value was obtained from Kruskal–Wallis test. **d** Coimmunofluorescent labelling of GFP (green) with CYP21 (purple) or DAB2 (blue) on WT, *Senp2^cKO*, *Prkar1a^cKO* or double

knockout adrenals. **e** Plasmatic concentration of corticosterone in 4-week-old WT, *Senp2^cKO*, *Prkar1a^cKO* or double knockout male and female mice. *P* values were obtained from one-way ANOVA (male) or Kruskal–Wallis test (female). **f** Plasmatic concentration of ACTH in 4-week-old WT, *Senp2^cKO*, *Prkar1a^cKO* or double knockout male and female mice. *P* values were obtained from Kruskal–Wallis test and adjusted with FDR method. **g** PKA kinase activity measurements in WT and *Senp2^cKO* 4-week-old adrenals in presence or absence of 5 μM SUMO vinyl sufone. *P* values were obtained from Kruskal–Wallis test and adjusted with FDR method. **h** Western blot analysis of global SUMOylation and PKA catalytic subunits protein accumulation 4-week-old adrenals from WT and *Senp2^cKO* mice. *P value < 0.05; P value < 0.01; ***P value < 0.001. Source data are provided as a Source Data file.

## Table 1 | Mean concentration (± SD) of plasmatic steroids in 24-week-old WT and *Senp2^cKO* mice

| Male | Vehicle | | ACTH | |
|---|---|---|---|---|
| | WT | *Senp2^cKO* | WT | *Senp2^cKO* |
| Progesterone | 0.15 +/− 0.09 | 0.30 +/− 0.39 | 9.23 +/− 4.03 (0.0001) | 1.70 +/− 0.88 (0.0457) |
| DOC | 0.31 +/− 0.29 | 0.43 +/− 0.17 | 24.02 +/− 18.9 (0.0001) | 5.26 +/− 3.53 (0.0401) |
| Corticosterone | 26.6 +/− 13.89 | 24.39 +/− 13.36 | 202.3 +/− 40.96 (0.0001) | 75.53 +/− 22.10 (0.0267) |
| 11-DH-cort | 0.13 +/− 0.11 | 0.12 +/− 0.06 | 0.89 +/− 0.68 (0.0004) | 0.18 +/− 0.06 (0.3815) |
| **Female** | **Vehicle** | | **ACTH** | |
| | WT | *Senp2^cKO* | WT | *Senp2^cKO* |
| Progesterone | 7.31 +/− 15.52 | 5.56 +/− 11.57 | 3.55 +/− 1.62 (0.9999) | 2.60 +/− 1.08 (0.2182) |
| DOC | 4.91 +/− 4.20 | 9.65 +/− 11.53 | 55.75 +/− 14.29 (0.0001) | 28.02 +/− 13.62 (0.0896) |
| Corticosterone | 106.70 +/− 65.69 | 112.80 +/− 47.87 | 334.20 +/− 67.24 (0.0001) | 200.00 +/− 81.52 (0.0113) |
| 11-DH-cort | 0.49 +/− 0.33 | 0.45 +/− 0.18 | 2.30 +/− 1.01 (0.0001) | 0.85 +/− 0.36 (0.1544) |

Plasmatic concentrations were determined by LC–MS/MS. *P* values (in brackets) compared to vehicle counterpart was obtained by Kruskal–Wallis with multiple comparisons using Dunn's test.

the major PKA catalytic subunits Cα and Cβ. We found a conserved SUMO consensus motif on the Lys169 encoded by the 6th exon (Fig. 4a). In vitro SUMOylation assay revealed SUMOylated forms of both C subunits detectable only when ATP was added to the SUMO machinery. Importantly, these SUMO conjugates disappeared after addition of SENP2 (Fig. 4b) consistent with the interaction between SENP2 and Cαβ. Moreover, proximity ligation assay (PLA) (which enables to visualise in situ the close proximity of two proteins) revealed SUMOylated catalytic subunits in approximately 40% of WT adrenal cortical cells whereas more than 95% harboured SUMOylation *foci* in *Senp2^cKO* cortices suggesting increased in vivo SUMO conjugation of PKA following SENP2 deficiency (Fig. 4c). In conclusion, SENP2 has the capacity to deSUMOylate PKA catalytic subunits α and β which is probably the cause of their hyperSUMOylation in *Senp2^cKO* adrenals.

To test the involvement of SUMOylation in the blunted response to ACTH after SENP2 loss, we treated *Senp2^cKO* mice and their WT littermates with SAE1/UBA2 inhibitor TAK-981[30] or vehicle between 3 and 8 weeks of age (Fig. 4d). Monitoring the weight growth of mice from 3 weeks of age showed that *Senp2^cKO* mice had a lower mass than WT between 6 and 7 weeks, consistent with weight loss observed in patients suffering from adrenal insufficiency[31] (Fig. 4e). TAK-981 treatment led to normalisation of this phenotype, indeed, body mass of WT and *Senp2^cKO* were comparable in treated mice (Fig. 4f). To explore the effects of pharmacological attenuation of SUMOylation on adrenal cortex responsiveness, we challenged the mice with ACTH injection 30 min prior to harvest and measured corticosterone plasma levels. TAK-981 treatment induced an increase in corticosterone response to ACTH in WT mice and a tendency toward recovery in *Senp2^cKO* (Fig. 4g).

Taken together, these results show that while SENP2 loss is associated with decreased response to ACTH and higher PKA catalytic subunits SUMOylation, pharmacological attenuation of SUMOylation restores normal body mass and tends to improve response to ACTH in

*Senp2^cKO*, supporting the involvement of SENP2 in preventing SUMO-dependent repression of PKA activity.

### Senp2 is necessary for the acquisition of zF identity

Based on the blunted response to ACTH, we decided to assess the differentiation status of the zF in *Senp2^cKO* mice. We took advantage of cortical cells' capacity to escape recombination to compare the intensity of differentiation markers in neighbouring cells, differing only by their recombination status. We performed triple staining for GFP, used as a proxy of *Senp2* recombination, with DAB2 and AKR1B7 labelling zG and zF, respectively. We observed a consistent lower AKR1B7 staining intensity in GFP-positive cells, indicating that loss of *Senp2* hinders cells from expressing zF markers compared to neighbouring GFP-negative cells (Fig. 5a, b and Supplementary Fig. S5.a). RT-qPCR analysis revealed an increased accumulation of progenitors' markers and higher number of NR2F2-positive capsular cells in males (Supplementary Fig. S5b, c) implying a default in cortical cell turnover which, together with the downregulated AKR1B7 expression, suggests a block in centripetal differentiation. This hypothesis was further supported by an increased proportion of cells coexpressing DAB2 and AKR1B7 in the cortex of *Senp2^cKO*, suggesting altered zG-to-zF trans-differentiation (Fig. 5c). To examine this hypothesis, we performed functional lineage tracing analysis of *Senp2*-deficient cells using mTmG reporter mice and *AS^Cre* driver[2], which allowed to delete *Senp2* in zG cells after birth (*AS^Cre/+::Senp2^fl/fl::R26R^mTmG/mTmG*). This Cre driver enabled to assess cortex cellular turnover through the percentage of GFP-stained cells progressing centripetally. As previously shown[7,32], complete cortical cell renewal took around 12 weeks in female and 40 weeks in male WT mice (Fig. 5d). Interestingly, whereas GFP immunostaining marked the first third of the cortex (zG and upper zF) in 4 weeks WT females, GFP staining was confined to the zG and some rare stripes projecting into the zF in *AS^Cre/+ Senp2^cKO* littermates (Fig. 5e).

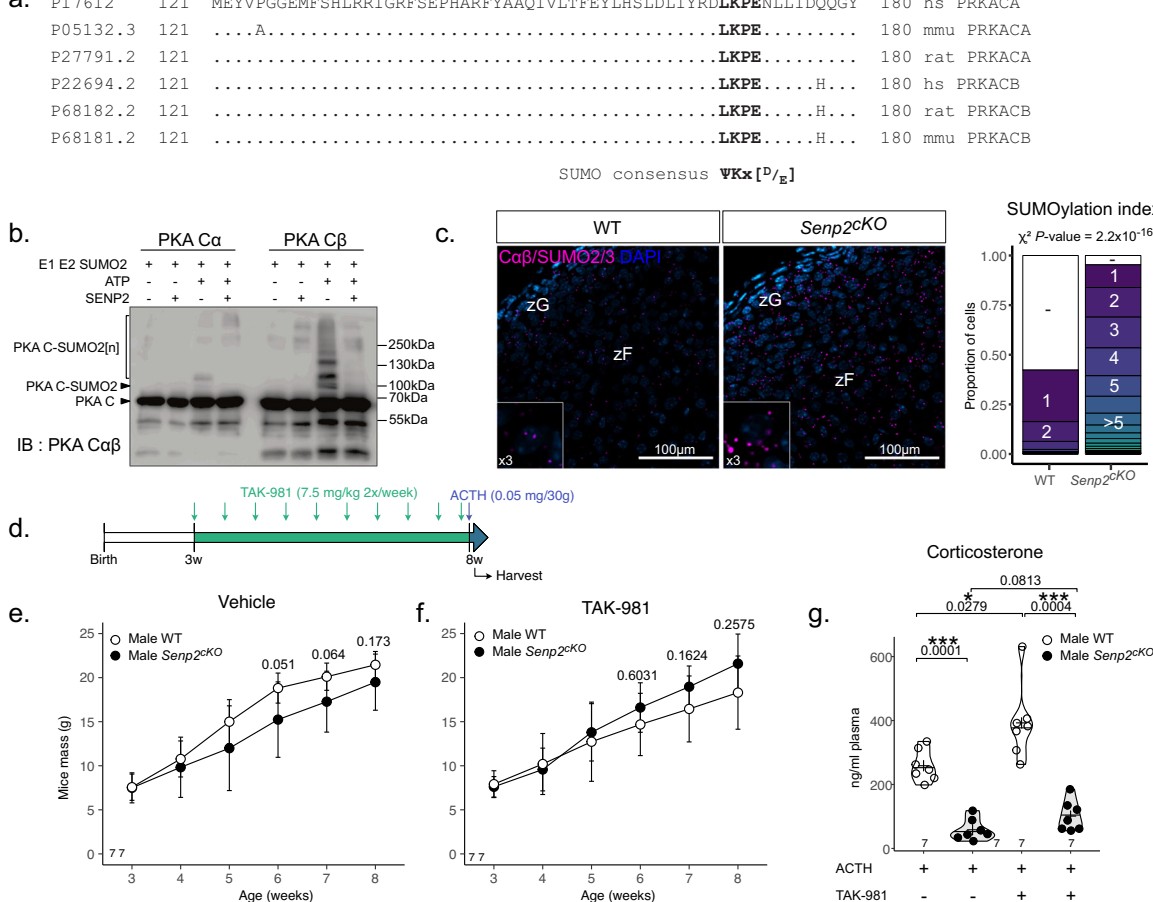

**Fig. 4 | SENP2 deSUMOylates PKA catalytic units. a** Sequence alignment of *Homo sapiens, Mus musculus* and *Rattus norvegicus* PRKACA (Cα) and PRKACB (Cβ). **b** In vitro SUMOylation assay of PKA Cα and Cβ in presence or absence of SENP2. **c** Proximity ligation assay analysis of PKA Cα/β in WT and *Senp2cKO* 4-week-old adrenals. **d** Schematic representation of the experimental setup of mice treatments. **e** Weight monitoring of WT and *Senp2cKO* mice treated with vehicle control between 3 and 8 weeks (duration 5 weeks). (*n* = 7 animals per condition). Data are presented as mean values +/− SD. **f** Weight monitoring of WT and *Senp2cKO* mice treated with TAK-981 between 3 and 8 weeks (duration 5 weeks). (*n* = 7 animals per condition). Data are presented as mean values +/− SD. **g** Corticosterone plasmatic concentrations in WT and *Senp2cKO* mice after 5 weeks vehicle (−) or TAK-981 treatments and 30 min after ACTH injection. *P* values were obtained from one-way ANOVA and adjusted with FDR method. *$P$ value < 0.05; **$P$ value < 0.01; ***$P$ value < 0.001. Source data are provided as a Source Data file.

To accelerate lineage tracing, we mated Cre homozygosity, and thus deleted the Aldosterone Synthase gene (*AS^Cre/Cre^::Senp2^fl/fl^::R26R^mTmG/mTmG^*). Consistent with previous reports[2], this enhanced trophic drive through renin-angiotensin signalling, and increased recombination rate, leading to almost full recombination cortex by 4 weeks of age in *AS^Cre/Cre^ Senp2* heterozygous female adrenal. In sharp contrast, GFP staining was almost confined to the zG in *AS^Cre/Cre^ Senp2^cKO^* even though the zG was slightly expanded because of trophic stimulation by angiotensin (Fig. 5e). A similar phenotype was observed in *AS^Cre/+^ Senp2^cKO^* at 40 weeks of age with a recombined zF consisting of scattered stripes of GFP-positive cells (Fig. 5e). We next quantified AKR1B7 protein accumulation in zF cells of *AS^Cre/+^ Senp2^cKO^* of 24 weeks of age and again observed lower AKR1B7 staining in GFP-positive than in GFP-negative cells (GFP- 95% CI [6887; 7164] AU, GFP + 95% CI [5427; 5582] AU, *P* value = 10⁻¹⁶) (Fig. 5f, g). In contrast with *Sf1-Cre* mediated inactivation of *Senp2*, *AS^Cre^* mediated deletion did not alter adrenal weight at 24 weeks of age (Fig. 5h). However, when ACTH responsiveness was assessed over time, plasma corticosterone peaked 2 h after ACTH treatment in WT, whereas the response was slower in *AS^Cre/+^ Senp2^cKO^* and never reached statistical threshold (Fig. 5i). This shows that even in the absence of adrenal hypoplasia, *Senp2* ablation results in a block of zF transdifferentiation from zG cells. This causes incomplete zF differentiation in recombined cells and allows competitive

selection of non-recombined cells, which cannot completely overcome the endocrine phenotype.

### Senp2-deficient cells undergo apoptosis associated with DRP1 phosphorylation

We next examined the cellular mechanisms underlying the development of zF atrophy in *Senp2^cKO^* mice, by analysing the proliferation/apoptosis balance. The cortical proliferation index determined by scoring the number of Ki67- or BrdU-positive cells ruled out the contribution of a decreased proliferation rate to the hypoplastic phenotype, but rather showed a trend toward increased cell division in mutant adrenals (Supplementary Fig. S6a, b). Nonetheless, cleaved caspase-3 staining showed that the numbers of cells undergoing apoptosis was dramatically increased in *Senp2^cKO^* adrenals at 4 and 8 weeks of age in both sexes (Fig. 6a). Whereas, according to the standard model, apoptosis is normally found at the corticomedullary junction (where adrenal cells die after centripetal migration[33]), in the *Senp2^cKO^* cortex, apoptosis occurred prematurely at the border between the zG and zF (Fig. 6a). DRP1 (Dynamin Related Protein 1) is considered the primary driver of mitochondrial fission and mitochondrial-dependant cell death[34,35]. Phosphorylation of DRP1 on Ser616 activates mitochondrial fission while that on Ser637 prevents the fission. The dysregulation of DRP1 phosphorylations on these two

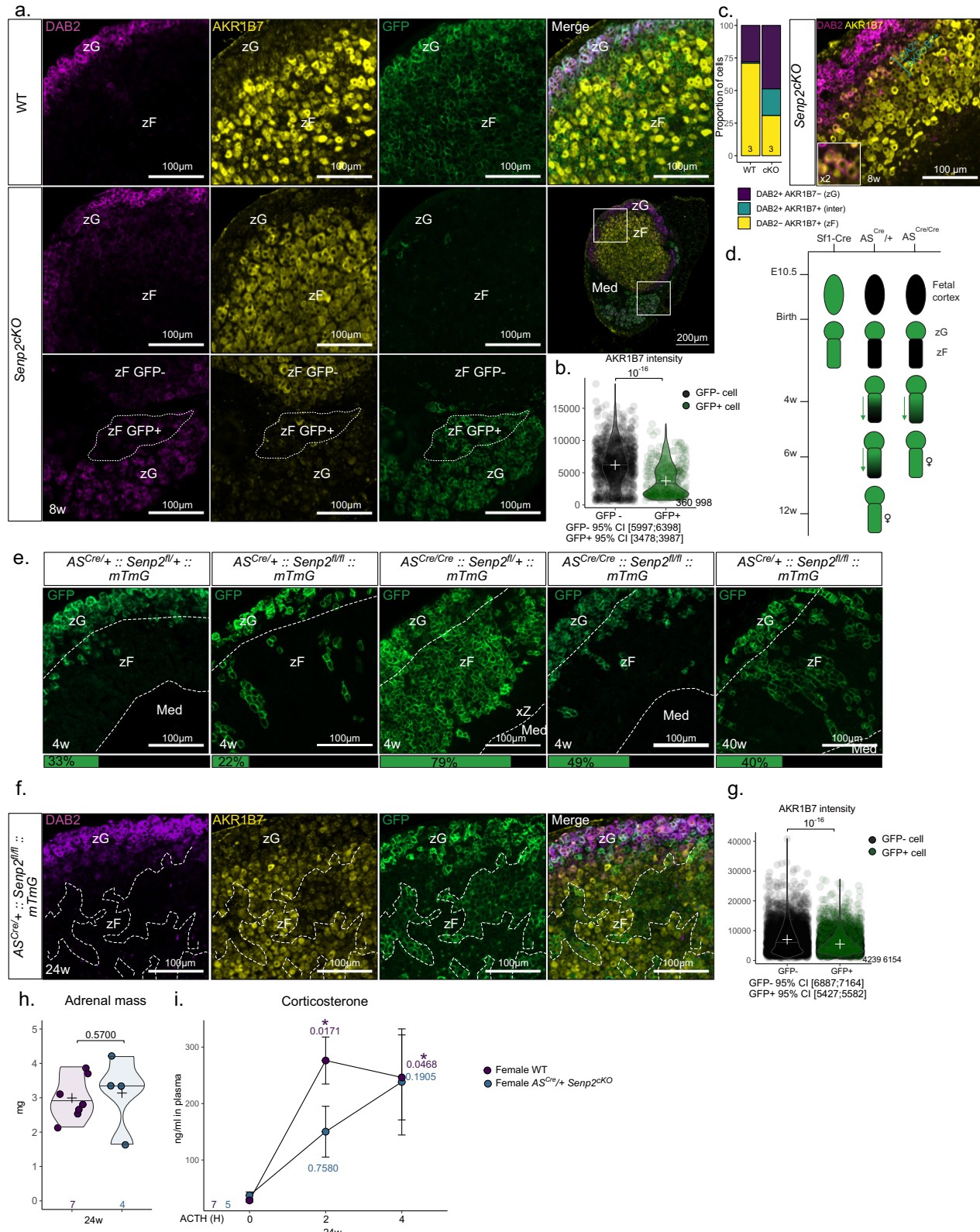

residues will result in imbalanced mitochondrial fission/fusion, a major cause of apoptotic cell death. Phosphorylation on Ser637 particularly raised our attention since it is catalysed by PKA[36]. Besides its participation to inhibition of mitochondrial fission, studies have shown that phosphorylated DRP1 Ser637 promotes steroidogenesis in Leydig cells[37] and *corpus luteum*[38]. To assess the impact of ACTH/PKA on DRP1 activity in adrenal glands, we performed western blotting of DRP1 and

its Ser637 phosphorylated form in WT or *Senp2^cKO* adrenal treated with vehicle or ACTH for 30 min. We found no difference in total or phosphorylated DRP1 in vehicle treated mice. In contrast, ACTH treatment induced DRP1 Ser637 phosphorylation solely in WT mice whereas this response was abolished in *Senp2^cKO* adrenals (Fig. 6b).

We then took an in vitro approach to determine whether PKA-induced changes in DRP1 phosphorylation were involved in

**Fig. 5 | Senp2 ablation prevents proper zF differentiation.**
**a** Coimmunofluorescence labelling of AKR1B7 (yellow), GFP (green) and Disabled2/
DAB2 (purple) on 8-week-old WT and *Senp2ᵏᴼ* male adrenals. **b** Quantification of
AKR1B7 intensity in GFP+ and GFP− *Senp2ᵏᴼ* male adrenal cells. *P* values was
determined by two-sided Mann–Whitney test. **c** Quantification and representative
image of cells expressing DAB2 (purple), AKR1B7 (yellow) or both (blue) in WT and
*Senp2ᵏᴼ* 8-week-old adrenals. **d** Scheme representing the differences in recombi-
nation kinetics between cre drivers used in genetic models. **e** Immunofluorescence
labelling with mean percentage of GFP-positive cells in 4w and 40w cortex.
**f** Coimmunofluorescence labelling of AKR1B7 (yellow), GFP (green) and Disabled2/
DAB2 (purple) on female 24-week-old *ASᶜʳᵉ/+ Senp2ᵏᴼ* adrenals. **g** Quantification of

AKR1B7 intensity in GFP+ and GFP− 24-week-old *ASᶜʳᵉ/+ Senp2ᵏᴼ* adrenal cells. *P*
values was determined by two-sided Mann–Whitney test. **h** Mean adrenal mass of
24-week-old female *ASᶜʳᵉ/+ Senp2ᵏᴼ* and WT. *P* values was determined by two-sided
Mann–Whitney test. **i** Kinetics of ACTH response of 24-week-old female *ASᶜʳᵉ/+
Senp2ᵏᴼ* and WT. *P* values represent difference between samples from the same
mice before treatment compared to after 2 h or after 4 h of treatment. (*n* = 7 mice
for WT and 4 mice for *ASᶜʳᵉ/+ Senp2ᵏᴼ*). Data are presented as mean values +/− SD.
Corticosterone response to ACTH was analysed with two-sided paired two-way
ANOVA to compare the effect of treatment for each genotype and adjusted with
FDR method. *\**P* value < 0.05; **\*\**P* value < 0.01; *\*\*\**P* value < 0.001. Source data are
provided as a Source Data file.

the apoptotic response of adrenocortical cells. The *fasciculata*-
like ATC7 cells[39] were treated with forskolin (FSK), a pharmaco-
logical activator of PKA (through increased cAMP cellular levels),
1 h before a 2-h incubation with the proapoptotic drug staur-
osporine (STS). As expected, FSK alone resulted in increased
Ser637 phosphorylation of DRP1, while STS alone induced
caspase-3 cleavage. When combined, FSK limited STS-driven
apoptosis, as depicted by the reduced accumulation of cleaved
caspase-3 (Fig. 6c). Of note, Ser616 proapoptotic phosphory-
lation of DRP1 showed a trend to be regulated in the exact opposite way
to Ser637 (Fig. 6c). To specifically assess the contribution of DRP1
in STS-induced apoptosis, we pretreated cells with the DRP1-
specific inhibitor (Mdivi-1) before inducing apoptosis with STS[40].
Similar to FSK, pretreatment with Mdivi-1 resulted in protection
against apoptosis as shown by reduced accumulation of cleaved
caspase-3 (Supplementary Fig. S6c). Consistently in vivo, we
observed an inverse correlation ($_\tau$ = −0.3823, *P* value = 0.0341)
between cortical apoptosis rate and DRP1 Ser637 phosphorylation
in *Senp2*, *Prkar1a* single and double knockouts (Fig. 6d–f and
Supplementary Fig. S6d). Altogether, these results strongly sug-
gest that increased apoptosis seen at the zG-zF boundary in
*Senp2ᵏᴼ* adrenals results from a deficient ability of ACTH/PKA
signalling to properly phosphorylate DRP1 Ser637.

### *Senp2* deficiency leads to β-catenin hyperSUMOylation and mild activation of WNT pathway

To unravel new SUMOylation-sensitive pathways that could explain
further the adrenal insufficiency of mice lacking SUMO protease
SENP2, we performed bulk RNA sequencing on four-week-old male and
female WT and *Senp2ᵏᴼ*. We found 1337 genes to be differentially
expressed in male (1115 up and 222 down) and 1235 in female (960 up
and 275 down) (Fig. 7a, b). Unsupervised clustering and principal
component analysis discriminated samples based on genotype but not
on sex (Supplementary Fig. S7a, b), implying that at 4 weeks of age, sex
has a low impact on gene transcription. Genomic alignments of reads
confirmed ablation of exon 13 and 14 of SENP2 in mutant adrenals
(Supplementary Fig. S7c). Since most of the genes were co-regulated in
males and females, we chose to focus on these subsets of genes. We
performed Gene Ontology (GO) functional enrichment analysis on
upregulated or downregulated genes in *Senp2ᵏᴼ* adrenals of both
sexes (Fig. 7c). The top GO terms associated with the upregulated
genes were linked to neuron cells and function, which may be due to
over-representation of medullar chromaffin cells resulting from cor-
tical hypoplasia. Pathways related to steroid processing were enriched
in downregulated genes, consistent with the endocrine deficiency
phenotype of *Senp2ᵏᴼ* mice.

We performed Gene Set Enrichment Analysis (GSEA) on the
Kyoto Encyclopedia of Genes and Genomes (KEGG) database and
selected only the enriched pathways that were present in both
males and females (Fig. 7d). This confirmed negative enrichment
of the gene signature for steroid hormone biosynthesis pathway,
in accordance with GO analysis and hormonal insufficiency char-
acterised in *Senp2ᵏᴼ* mice (Fig. 2). Moreover, we found negative

enrichment of signatures associated with nucleotide excision
repair, aminoacyl tRNA biosynthesis, ribosome, pyrimidine
metabolism and spliceosome, indicating that *Senp2* loss in the
adrenal cortex altered basic cellular processes known to be
regulated by SUMOylation[12]. Interestingly, among the positively
enriched pathways, WNT signalling caught our attention as it is
mandatory for adrenocortical maintenance and proper
zonation[41–43] (Fig. 7d, e). Based on RNA-Seq analyses and given
that SENP2 has first been described as a negative regulator of β-
catenin[44–46], we sought out to see if the WNT/β-catenin pathway
was altered in *Senp2ᵏᴼ* adrenals. We evaluated the activation of
the pathway by running RT-qPCR on its target genes. At 24 weeks
of age, we observed a consistent ~1.5-fold induction of *Axin2*, *Lef1*,
*Apcdd1* and *Ccdc80* in the adrenals of *Senp2ᵏᴼ* females, whereas in
mutant males, only *Ccdc80* was upregulated by about threefold
(Fig. 8a). To gain more insight into the modulation of the WNT
signalling pathway, we extracted expression levels of 48 WNT
target genes from RNA-seq data (Fig. 8b). Among these genes, 31
were upregulated mainly in males *Senp2ᵏᴼ* and 7 mainly in
females while 10 genes were downregulated in both sexes.

We next carried out immunostaining on adrenal sections that
revealed a nuclear localisation of non-phospho (active) β-catenin
specifically in the inner cortex of mutant adrenals regardless of sex
(Fig. 8c, top). Co-staining of total β-catenin with zF marker AKR1B7
confirmed zF identity of these cells, which were also GFP-positive
and hence, had been targeted by Cre recombination (Fig. 8c,
bottom).

Given that β-catenin can be SUMOylated by both SUMO1[47] and
SUMO2/3[48], we hypothesised that it was SUMOylated in the adrenal
cortex. We performed immunoprecipitation of both SUMO1 or
SUMO2/3 and blotted β-catenin. We observed a doublet between 120
and 150 kDa for SUMO2/3 consistent with a previous report[48] but no
evidence for significant SUMO1 conjugation (Fig. 8d). To evaluate if β-
catenin SUMO2/3ylation was affected upon *Senp2* deletion, we
immunoprecipitated β-catenin and blotted SUMO2/3. Although we did
not see any difference in the amount of SUMOylated β-catenin, the
accumulation of its native form was reduced in mutant adrenals,
whereas global SUMOylation by SUMO2/3 was unchanged (Fig. 8e).
Reciprocally, by immunoprecipitating SUMO2/3 and blotting β-cate-
nin, we observed a slight but reproducible accumulation of SUMOy-
lated β-catenin compared to native form (Fig. 8f). To confirm this
increase in β-catenin SUMO2/3ylation, we used Proximity Ligation
Assay (PLA) (Fig. 8g). As a negative control, we first ran PLA between β-
catenin and GATA6, a transcription factor present in the nucleus of all
adrenocortical cells and which is not known to interact with β-catenin.
As expected, there were no *foci* of β-catenin/GATA6 interaction in the
adrenal sections of all genotypes. In contrast, PLA between β-catenin
and SUMO2/3 showed specific *foci* in the zG with similar density in
both genotypes. However, in the zF of *Senp2ᵏᴼ*, there was an increase
in cells containing 2 or more β-catenin/SUMO2/3 dots per
nucleus (Fig. 8g).

In conclusion, loss of SENP2 leads to ectopic nuclear accumula-
tion of β-catenin in the zF associated with its increased conjugation to

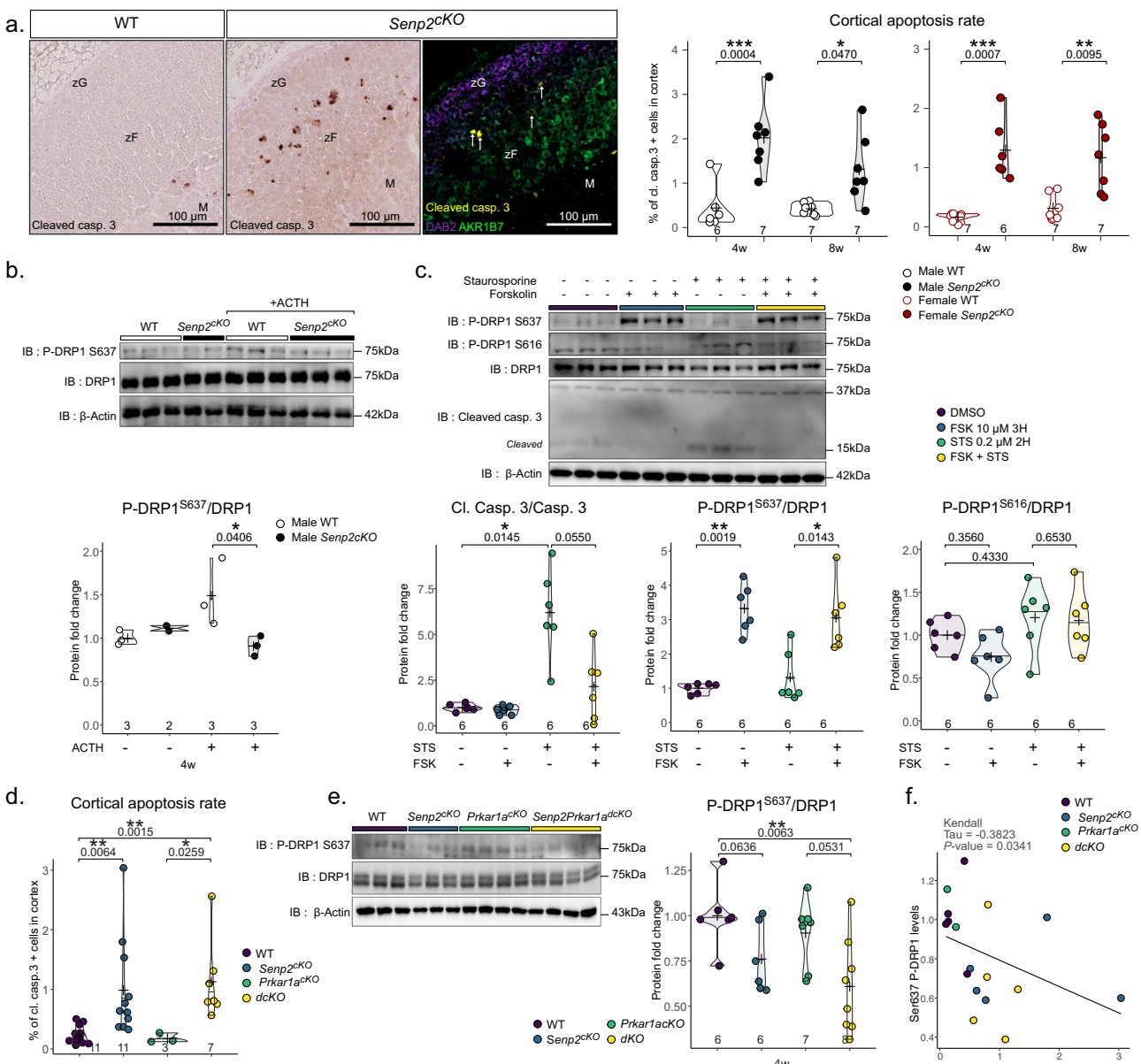

**Fig. 6 | *Senp2* loss triggers apoptosis at the zG-zF boundary. a** Representative image of cleaved caspase-3 immunostaining and immunofluorescence labelling of cleaved caspase-3 (yellow), zG marker Disabled2/DAB2 (purple) and zF marker AKR1B7 (green). Quantification of apoptosis in WT and cKO through measurement of the percentage of cells positive for cleaved caspase-3 at 4 and 8 weeks of age. *P* values were determined by a two-sided *t* test for normally distributed condition or two-sided Mann–Whitney test. **b** Western blot analysis of S637 phosphorylated and total DRP1 in adrenal after 30 min ACTH I.P. treatment of WT and *Senp2cKO* male mice. *P* value was obtained from Kruskal–Wallis test. **c** Western blot analysis of phosphorylated and total DRP1 in cells treated with DMSO, Forskolin (10 μM) and/or Staurosporine (0.2 μM). *P* value was obtained from Kruskal–Wallis test and adjusted

with FDR method. **d** Quantification of cleaved caspase-3 positive cells in the cortex (**e**) of WT, *Senp2cKO*, *Prkar1acKO* and double- knockout male adrenals. *P* value was obtained from Kruskal–Wallis test and adjusted with FDR method. **e** Western blot analysis of S637 phosphorylated and total DRP1 in the cortex (**e**) of WT, *Senp2cKO*, *Prkar1acKO* and double-knockout males adrenals. *P* value was obtained from Kruskal–Wallis test and adjusted with FDR method. **f** Correlation plot between Ser637 DRP1 phosphorylation and proportion of cleaved caspase-3-positive cells in the adrenal cortex across genotypes. Correlation has been conducted using Kendall method as distributions did not follow normality and contained ties. Source data are provided as a Source Data file. *P value < 0.05; **P value < 0.01; ***P value < 0.001.

SUMO2/3. This accumulation is correlated with a mild activation of WNT pathway targeting genes in a partially sexually dimorphic pattern.

## Discussion

We have previously established that the overall SUMOylation of proteins in the adrenal cortex displays two remarkable properties: it gradually decreases as the centripetal transdifferentiation of zG cells into zF cells progresses and it is acutely downregulated by stress through ACTH/PKA signalling[20]. The present study was initiated to address the

hypothesis that this posttranslational mechanism involved in the cellular response to various environmental stressors[49] may be an integral part of the adrenal gland's toolbox to produce steroid hormones critical for body homoeostasis and stress adaptation. Adrenal-specific KO mouse models of SENP2 deSUMOylase and of PKA regulators established that this is indeed the case. Foetal or postnatal SENP2-dependent deSUMOylation is necessary for initial zF differentiation, its maintenance throughout life and for ACTH/PKA-stimulated glucocorticoid production. As a result, foetal or postnatal deletion of *Senp2*

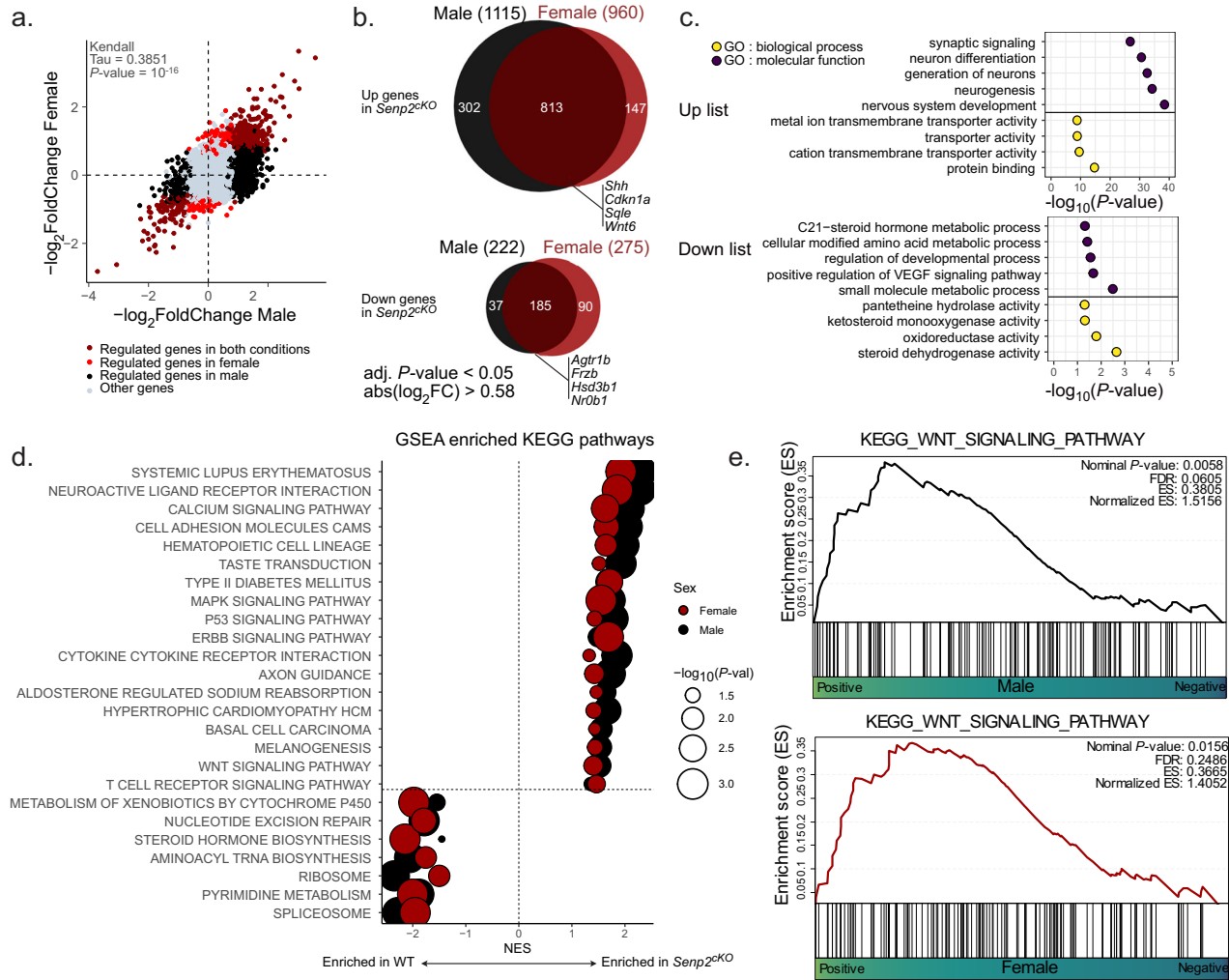

**Fig. 7 | RNA-seq analysis of male and female *Senp2*<sup>cKO</sup> adrenals. a** Scatter plot showing the correlation between dysregulated genes (adjusted *P* value <0.05 and absolute log₂ fold change > 0.58) in male and female *Senp2*<sup>cKO</sup> adrenals at 4 weeks of age. Correlation has been conducted using Kendall method as distributions did not follow normality and contained ties. **b** Euler diagrams illustrating the commonly up or downregulated genes in both sexes *Senp2*<sup>cKO</sup> adrenals at 4 weeks of age. **c** Top GO terms from Gene Ontology functional enrichment analysis based on the up- or downregulated genes lists. **d** Gene set enrichment analysis (GSEA) pathways from the KEGG database that are commonly affected in male and female *Senp2*<sup>cKO</sup> adrenals. **e** Gene set enrichment analysis (GSEA) plots of WNT pathway on male and female WT versus *Senp2*<sup>cKO</sup> adrenals. Source data are provided as a Source Data file. *P value < 0.05; **P value < 0.01; ***P value < 0.001.

in adrenal steroidogenic cells causes postnatal hypoplasia limited to the zF or incomplete zF formation respectively, which can lead to isolated glucocorticoid deficiency. This selective atrophy and associated endocrine deficit result both from a blockage of zonal trans-differentiation preventing the switch from zG to zF identity, and from a repression of ACTH/PKA responsiveness. Interestingly, overt corticosterone deficiency is only measurable in males at 4 weeks of age, even though ACTH levels are higher at all ages tested in both genders. This implies that a subclinical deficiency is permanently established in the mutants, despite setting up compensation. This compensation to maintain physiological plasma glucocorticoid levels, which occurs in both sexes but with delay in males, most likely relies on the emergence of a population of cells that has escaped recombination and helps the mutant cortex to overcome zF atrophy. Indeed, their hypertrophy indicates that they are overreacting to the high levels of circulating ACTH, to compensate for the lack of corticosterone due to the dramatic atrophy of the *Senp2*-deficient zF cell population.

We hypothesise that steroidogenic progenitor cells, which do not express the Cre recombinase for a still unknown reason, must gain a selective advantage over *Senp2*-deficient cells, either by being able to proliferate more or by being less prone to death upon differentiation.

Similar observations have been made in adrenals following SF-1 loss driven by *AS*<sup>Cre 2</sup> or *Cyp11a1-Cre*<sup>50</sup>, where a majority of cells escaped recombination. How cells manage to express a protein but not the Cre while both genes depend on the same promoter is still unknown. One possibility could be methylation of the promoter. For instance, *Ins1-Cre* transgene is shown to be a poor quality driver to target β cells in the pancreas as its expression is silenced by de novo methylation even though *Ins1* gene is normally expressed<sup>51</sup>. In any case, the triggering of a compensatory mechanism underlines the absolute necessity of maintaining *Senp2* expression (and thus the possibility of reducing SUMOylation), in order to build a functional *zona Fasciculata* to maintain the individual's ability to adapt to stress. The fact that *Senp2* deficiency preferentially affects the zF while being expressed in all cortical areas<sup>52</sup> can be explained by two non-mutually exclusive mechanisms. First, the zG already harbours high levels of nuclear SUMOylation compared to the zF<sup>20</sup>, therefore the expected increase in SUMOylation conjugates consecutive to lowered deconjugase pools will preferentially affect hypoSUMOylated regions. Moreover, we showed that SENP2 loss alters the PKA/WNT balance toward WNT signalling. The latter being already highly active in the zG<sup>4,6,41</sup>, it is not surprising to find an altered phenotype only in the zF where WNT/β-

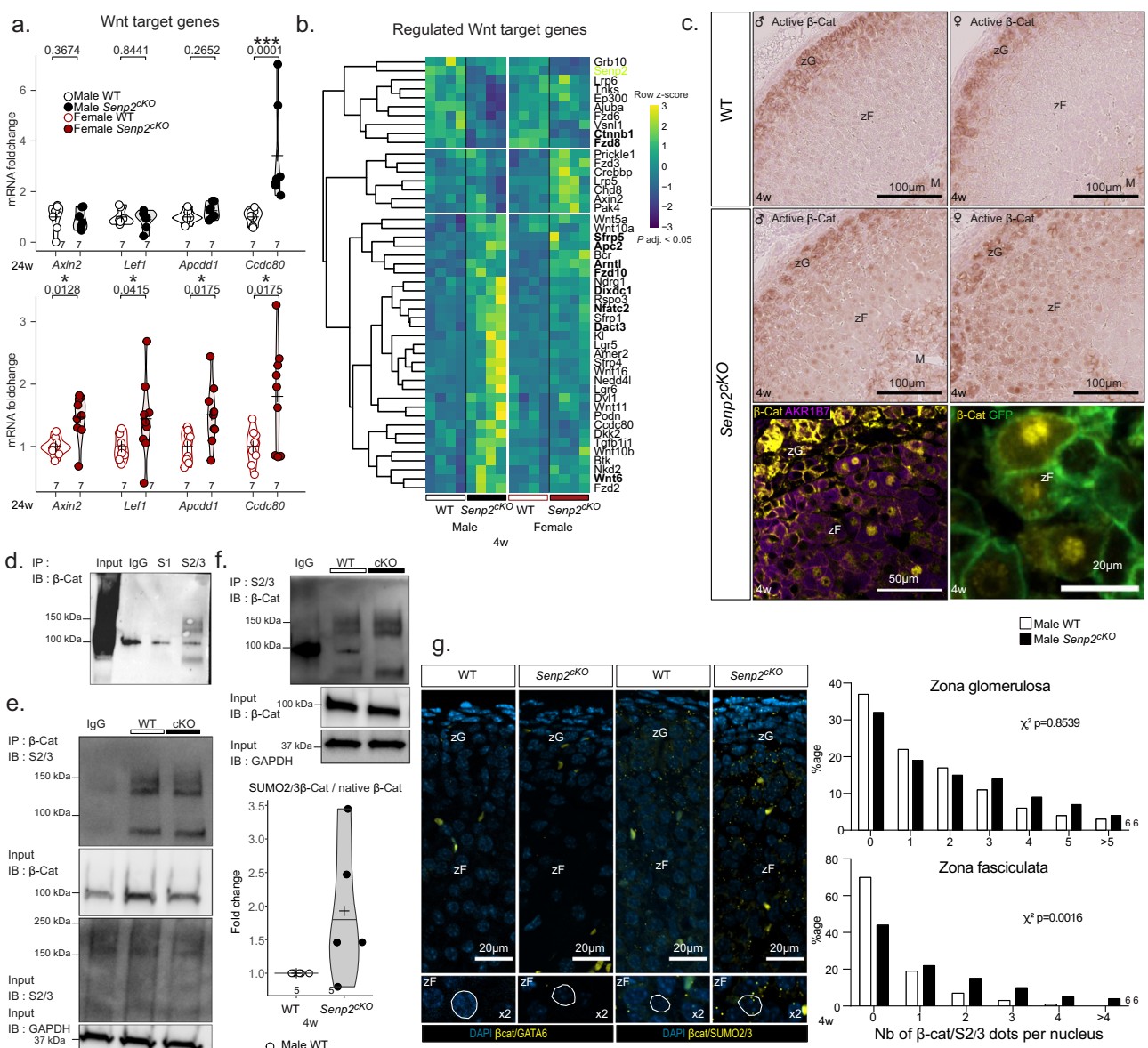

**Fig. 8 | _Senp2_ loss promotes β-catenin SUMOylation and activates the WNT signalling pathway. a** qPCR analysis of β-catenin target genes _Axin2_, _Lef1_, _Apcdd1_ and _Ccdc80_ mRNA accumulation in 24-week-old WT and _Senp2^{KO}_ mice. _P_ values were determined by two-sided _t_ test for normally distributed condition or two-sided Mann–Whitney test. **b** Heatmap of dysregulated WNT target genes in _Senp2^{KO}_ male or female adrenals RNA-seq (adjusted _P_ value < 0.05). Commonly dysregulated genes in males and females are represented in bold characters. **c** Top: Immunohistochemistry analysis of active (non-phospho) β-catenin on WT and _Senp2^{KO}_ male (left) and female (right) adrenals. Bottom: immunofluorescence staining of β-catenin (yellow) with zF marker AKR1B7 (purple) and GFP (green). **d** Immunoprecipitation assay depicting the interaction between β-catenin and

SUMO1 or SUMO2/3 in WT adrenals. **e**, **f** Immunoprecipitation assay depicting the interaction between β-catenin and SUMO2/3 in adrenals of 4-week-old WT and _Senp2^{KO}_ male mice. Extracts were immunoprecipitated with β-catenin (**e**) or SUMO2/3 antibodies (**f**). SUMOylated β-catenin was quantified relative to its native form. **g** Proximity ligation assay (PLA) of β-catenin and GATA6 (negative control) or β-catenin and SUMO2/3 in nuclei of 4-week-old WT and _Senp2^{KO}_ male mice's adrenals. Histograms represent proportion of cells in each zone containing the specified number of dots per nucleus of WT and _Senp2^{KO}_ male adrenals. _n_ = 6 per genotype. Ca capsule, zG _zona glomerulosa_, zF _zona fasciculata_. Source data are provided as a Source Data file. *_P_ value < 0.05; **_P_ value < 0.01; ***_P_ value < 0.001.

---

catenin signalling is naturally repressed[9]. The second reason is that low SUMOylation seems to be a prerequisite for adequate PKA signalling. Indeed, in our model, ACTH response is blunted and the zF atrophy phenotype is reminiscent, at least partially, of mouse models lacking ACTH receptor and its co-receptor i.e., _Mc2r^{−/−}_ or _Mrap^{−/−}_ whole-body knockouts[53,54]. Although various mechanisms could be simultaneously affected and remain to be explored, there are converging pieces of evidence that SENP2 deletion-dependant SUMOylation alters PKA responsiveness. Indeed, first, our data show that SENP2 deletion decreases overall PKA activity in the adrenal glands and prevents the

phosphorylation of specific targets (CREB Ser133, DRP1 Ser637, TRIM28 Ser473). Secondly, since _Senp2^{cKO}_ phenotype is not rescued by concomitant deletion of PKA regulatory subunit (_Prkar1a_), it is very likely that SENP2 loss directly affects the PKA catalytic subunits function. This is in line with various converging evidences (1) the capacity of catalytic subunits to be SUMOylated in vitro and deconjugated by SENP2 (Fig. 4a, b); (2) the higher SUMOylation of catalytic subunits in mutant cortices (Fig. 4c); (3) the improvement of adrenal ACTH responsiveness by selective SUMO E1 inhibitor (Fig. 4d–g); and (4) the decreased PKA activity caused by SUMO vinyl sulfone derivatives

(Fig. 3g, h). Further studies will assess the impact of SUMOylation of the catalytic subunits in fine-tuning PKA kinase activity.

SENP2 has first been described as a negative regulator of β-catenin stability[45] but this effect was independent of SUMOylation. Our model showed increased β-catenin SUMO2/3ylation along with ectopic nuclear β-catenin localisation associated with a gene signature indicative of the canonical WNT signalling. Similar observations have been reported following direct SUMOy2/3lation of β-catenin in mammary epithelial cells[48] or SUMOylation of TBL1/TBLR1 that in turn increased chromatin recruitment of β-catenin and its oncogenic activity in colon cancer[55]. However, the picture is not as clear since β-catenin SUMO2/3ylation can trigger its degradation in vascular smooth muscle cells[56] or prevent it in mammary epithelial cells[48]. Targeting gain-of-function mutation of β-catenin in aldosterone-synthase expressing cells caused hyperplasia by blocking *glomerulosa* cells transdifferentiation into *fasciculata* cells[3]. However, in contrast to *Senp2kO* mice, zF cells never became atrophic in β-catenin gain-of-function. It was thus proposed that when physiological transdifferentiation is unachievable, the zF can be maintained by an alternative cellular pathway involving progenitor cells that bypass the zG state[2]. Here, we report the occurrence of β-catenin SUMOylation associated with its nuclear translocation which takes place ectopically in *Senp2*-deficient zF. This leads to moderate WNT target genes activation and together with blunted PKA activity, impairs acquisition of *fasciculata* identity. In this context where zG-to-zF zonal transdifferentiation is impaired, the deficit in PKA signalling alters the phosphorylation profile of the major mitochondrial fission GTPase, DRP1, leading to premature apoptosis of the neoformed mutant *fasciculata* cells, reinforcing the atrophy of the zone over time. The block in zonal transdifferentiation induced by zG-targeted stabilised β-catenin has been ascribed to increased rosette formation (the basal cellular organisation of zG cells forming flower-like structures) and/or impaired rosette resolution (the moment when zG cells exit the rosette to form a single column of cells typical of zF)[57]. Conversely, adrenal cortex lacking *Senp2* do not exhibit zG hyperplasia and thus should not display enhanced rosette formation. Our data indicate that mechanisms by which zG undergoes resolution during normal homoeostatic turnover could be controlled by SUMOylation. Identifying what targets of *Senp2*-dependent SUMOylation directly contribute to zG resolution and whether this is coupled to zF transdifferentiation will require extensive studies.

SUMOylation has already proven a role in adrenocortical development with the unSUMOylatable *SF-1^2KR/2KR* model, which results in adrenal cells expressing gonadal markers[21]. In ovaries which share a common foetal origin with adrenals, SUMOylation by the E3 SUMO-ligase activity of TRIM28, is essential for maintenance of granulosa cell fate at the expense of Sertoli-like identity[58], whereas SENP1 presence in stromal cells is necessary for proper oocyte maturation and ovulation[59]. From a broader perspective, our study belongs to a growing corpus of evidence associating SUMOylation with coordination of differentiation in different tissues and cell types such as white and brown adipose tissue[15,17,60], induced pluripotent stem cells[16], or intestine[18]. Seemingly counter-intuitively, the induction of the adrenal phenotype of *Senp2zKO* mice does not rely on an overall increase in SUMO conjugates levels but rather on specific overSUMOylation of certain cell populations (Supplementary Fig. S1j, k) or substrates belonging to signalling pathways crucial for adrenal homoeostasis. In contrast, other substrates might be unaffected by the loss of SENP2, such as SF-1 for which no expected increase in SUMOylation could be recorded, whereas an increase in SUMO-sensitive target genes is detected[21] that would instead suggest hypoSUMOylation of SF-1 (Supplementary Fig. S8). These findings are consistent with recent studies exploring the consequences of increased SUMOylation in the uterine stroma (by *Senp1* deletion) or decreased SUMOylation (by

*Ubc9* haploinsufficiency) in the intestine of *Apc* mutant mice that showed strong phenotypes in the absence of global changes in visible SUMO conjugate levels[61,62]. This illustrates the great versatility of this posttranslational modification pathway. Changes in SUMOylation capacity can have specific effects in vivo, especially when targeted steps are catalysed by various members of the same enzymatic family, such as E3 SUMO ligases or SUMO-specific proteases, which can therefore differ in their substrate specificity. With this in mind, we speculate that specific over-SUMOylated substrates produced in *Senp2*-deficient adrenals should result from the primary loss of the SUMO protease. Their SUMOylation could be reinforced by the unrepressed (hypo-phosphorylated) E3 SUMO-ligase activity of TRIM28[63,64], secondary to the blunted ACTH/PKA signalling.

Overall, the present paper demonstrates that preventing deSUMOylation by adrenocortical-specific SENP2 ablation in mouse induces zF hypoplasia associated with increased premature apoptosis along the lineage conversion zG-zF process, ultimately resulting in a blockage of the physiological differentiation process, translating to isolated glucocorticoid deficiency. This is linked to dysregulation of WNT/PKA balance in favour of WNT signalling. This shift highlights the central role of SUMOylation in physiological processes such as differentiation, tissue maintenance, and stress response. Furthermore, our data could broaden the scope of expected impacts of SUMOylation alterations to endocrine pathogenesis. Although pathogenic associations have been made with alterations in SUMO enzymes or substrates, direct causal links to pathologies have very rarely been established[13,65]. Our work suggests that genetic alterations leading to excessive SUMOylation could be associated with isolated glucocorticoid deficiency in patients.

## Methods

### Ethical approval declarations

Mouse experiments were conducted according to French and European directives for the use and care of animals for research purposes and were approved by the Comité d'Éthique pour l'Expérimentation Animale en Auvergne (project agreement #211522019061912052883), C2EA-02, at Institut National de Recherche pour l'Agriculture, l'Alimentation et l'Environnement, Research Centre Clermont-Theix, France (C2E2A).

### Cell culture

Adrenocortical tumour cell line 7 (ATC7) cells were established from an adrenal tumour derived from a mouse expressing the Simian Virus 40 large T (SV40 T) antigen under the control of the aldo-keto reductase 1b7 (*Akr1b7*) gene promoter specific to the adrenal cortex[39,66]. Cells were cultured on poly-D-lysine-coated 10 cm cell culture dishes (Millipore Sigma, Burlington, MA) in a DMEM-F12 medium (Thermo Fisher Scientific, Waltham, MA) at 37 °C in the presence of 5% $CO_2$, insulin (10 mg/mL), transferrin (5.5 mg/mL), selenium (6.7 ng/mL) (Thermo Fisher Scientific), L-glutamine (2 mM), penicillin 0.1 U/mL), streptomycin (0.1 mg/mL), 2.5% horse serum and 2.5% foetal calf serum. Cells were seeded in 12-well plates and cultures to subconfluence and then starved by replacing medium by serum-free medium the day before the addition of forskolin (Sigma-Aldrich), staurosporine (Sigma-Aldrich) or Mdivi-1 (Merck) at the times and concentrations indicated in the figure's legends. Authentication of ATC7 line was performed in June 2021, testing ACTH ($10^{-8}$ M) and forskolin ($10^{-5}$ M) responsiveness of 24-h corticosterone production (ELISA) and 6-h induction *Mc2r*, *Scarb1*, *Star*, *Akr1b7* and *Cyp11b1* gene expression (RT-qPCR), respectively.

### Mice, hormonal measurements and TAK-981 treatment

Mice bred in-house and maintained on a mixed sv129-C57Bl/6 genetic background were housed on a 12-h light/12-h dark cycle (lights on at

7:00 am). Mice were fed normal, commercial rodent chow and provided with water *ad libitum*. After weaning, mice were kept in siblings with a maximum of four animals per cage.

At the end of experimental procedures mice were killed by decapitation around 8:30 am and trunk blood was collected in vacuum blood collection tubes (VF-053STK, Terumo). For ACTH treatments mice were injected intraperitoneally with vehicle control or 0.05 mg/30 g Synacthene (0.25 mg/mL, Novartis, Basel, Switzerland) 2 h or 30 min prior trunk blood sampling. ACTH response kinetics in Fig. 5i, was done with the collection of the blood from the tail of the mice at 8 am, 10 am and 12 pm.

For SAE1/UBA2 inhibitor treatments, *Senp2^{cKO}* (*Sf1-Cre/+::Senp2^{fl/fl}*) male mice and WT (*+/+::Senp2^{fl/fl}*) littermates aged 3 weeks were treated with vehicle control or TAK-981 (MedChemExpress, Sollentuna, Sweden) (7.5 mg/kg, intraperitoneally twice per week) for 5 weeks according to ref. 67. At the end of the 5 weeks treatment, mice received ACTH by intraperitoneal injection of 0.05 mg/30 g Synacthene (0.25 mg/mL, Novartis, Basel, Switzerland) 30 min prior being killed by decapitation.

Corticosterone was measured from plasma with ELISA kit (AR E-8100, LDN), ACTH was measured with Milliplex Map Kit (MPTMAG-49K, Millipore) and other steroids were measured by LC–MS/MS[68].

Humane endpoints resulting in immediate euthanasia included hunched posture, rough hair coat, signs of dehydration (reduced skin turgor, sunken eyes), abnormal respiration (tachypnea, dyspnoea, coughing), reduced or impaired mobility affecting the ability to obtain food or water, pallor or cyanosis, haemorrhage or bleeding from any orifice, diarrhoea, constipation or markedly reduced food intake; neurologic abnormalities (seizures, paralysis, circling, head tilt), impaired ability to urinate or defecate, visible jaundice, or loss of >15% normal body weight from pre-study baseline. CO2 euthanasia was performed according to the AVMA Guidelines for the Euthanasia of Animals (2020 Edition).

### Histology and proximity ligation assay

Adrenals were fixed in 4% PFA for 6 h and embedded in paraffin. In all, 5-μm sections were deparaffinised and processed for H&E. For immunohistochemistry or immunofluorescence, deparaffinised slides were submerged in antigen retrieval buffer and microwaved for 8 min.

After being rinsed with 1× PBS, they were blocked for an hour with 2.5% horse serum (Vector) and incubated overnight at 4 °C with primary antibody. After rinsing, they were incubated with ImmPRESS polymer for 30 min at room temperature. HRP activity was detected with NOVAred (Vector) or Alexafluor (Thermo Fisher). Primary antibodies are listed in Supplementary Table S2.

For PLA, blocked slides were incubated overnight at 4 °C with indicated antibodies followed by Duolink in situ PLA (Sigma-Aldrich) anti-mouse (minus) and anti-rabbit (plus) probes and detection reagents according to manufacturer's instructions.

Images were acquired with Zeiss Axioscan Z1 or Zeiss Imager M2 and ZEN 3.4 blue edition software and analysed with QuPath 0.3.2 software[69].

### Lipid droplet analysis on cryosections

For frozen sections, adrenal were fixed in 4% PFA for 6 h and immersed into 10% and 15% PBS-sucrose solutions for 20 min each, then 20% PBS-sucrose solution for 1 h, and in 50/50 OCT-Sucrose 20% solution overnight. They were subsequently placed in embedding moulds and with pure OCT and frozen to be stored at −80 °C.

To detect lipid droplets, 14-μm sections were cut from OCT-embedded adrenals. Sections were washed thrice in PBS 1x and incubated for 30 min in the dark with 10 μg/ml Bodipy 493/503 solution. After staining, sections were washed thrice in PBS 1× and mounted with VECTASHIELD Antifade Mounting Medium with DAPI (Vector) to stain nuclei.

### RT-qPCR

Adrenal glands were removed, flash frozen on dry ice, and RNA was extracted using RNeasy micro kit from QIAGEN. After reverse transcription, PCR reaction was conducted using SYBR qPCR Premix Ex Taq II Tli RNase H + (TAKRR820W, Takara). Primer pairs are listed in Supplementary Table S1.

### RNA-Seq

For each sex, adrenal gene expression profiles for four 4-week-old *Sf1-Cre/+::Senp2^{fl/fl}* and four WT littermates were analysed using RNA-seq. RNA sequencing, library generation and differential expression genes analysis were performed by the GenomEast platform (IGBMC, Illkirch, France).

Image analysis and base calling were performed using RTA 2.7.3 and bcl2fastq 2.17.1.14. Adapter dimer reads were removed using DimerRemover. Reads were mapped onto the mm10 assembly of *Mus musculus* genome using STAR version 2.5.3a. Gene expression quantification was performed from uniquely aligned reads using htseq-count version 0.6.1p1. Read counts have been normalised across samples with the median-of-ratios method proposed by Anders and Huber. Differential expressions have been implemented using the Bioconductor package DESeq2 version 1.16.1. Raw and processed data have been deposited in NCBI's GEO database (GSE193480).

Gene Set Enrichment Analysis was performed using the GSEA software[70] and plotted using the replotGSEA function from the Rtoolbox package (https://github.com/PeeperLab/ Rtoolbox). Gene ontology analysis was performed using g:Profiler[71]. Data visualisation was carried out using R software (v4.1.0)[72], Pheatmap package was used for heatmaps, Vennerable for Euler diagrams and ggplot2 for plots. PCA analysis was produced on read counts matrix using prcomp function from the stats package and plotted using ggplot2.

### Western blot and immunoprecipitation

Proteins were extracted from snap-frozen adrenals in RIPA buffer (TRIS 25 mM, EDTA 1 mM, MgCl$_2$ 5 mM, NP40 1%, glycerol 10%, NaCl 150 mM supplemented extemporaneously with phosphatase inhibitors (1 mM Na$_3$VO$_4$, 0.5 mM NaF), protease inhibitors (Roche, Basel, Switzerland), and SUMO proteases inhibitor N-ethylmaleimide (Millipore Sigma) (3.13 mg/mL).

For western blot, 30 μg of total denatured proteins were loaded on 10% SDS-page gel, transferred on nitrocellulose and detected with primary antibodies (Supplementary Table S2). Signals were quantified with ChemiDoc MP Imaging System camera system (Bio-Rad) and Image Lab software (Bio-Rad). Expression of phosphorylated or SUMOylated proteins was normalised to the expression of the corresponding unmodified protein.

Immunoprecipitation was carried out using Automag Solution (AdemTech). In total, 500–1000 μg of total proteins were precleared with 50 μL of beads for 30 min at room temperature. In all, 10 μL of antibodies (Supplementary Table S2) were crosslinked with 50 μL of beads in 20 mM DMP for 30 min. Precleared samples were immunoprecipitated with crosslinked antibodies for 60 min at room temperature, washed thrice in RIPA buffer and eluted with 50 μL of 50 mM glycine pH 3. pH was brought back to neutral with 1 μL of TRIS buffer pH9 and samples were denatured in Laemmli buffer (Bio-Rad) at 95 °C for 5 min and loaded for SDS-PAGE.

### PKA activity

Proteins were extracted from snap-frozen adrenals in lysis buffer (MOPS 20 mM, betaglycerol-phosphate 50 mM, NP40 1%, DTT 1 mM, EDTA 2 mM, EGTA 5 mM supplemented extemporaneously with phosphatase inhibitors (1 mM Na$_3$VO$_4$, 50 mM NaF), protease inhibitors (Roche, Basel, Switzerland). HA-SUMO vinyl sulfone (R&D Systems, Minneapolis, MN) was added for the SUMO-VS condition to the concentration of 5 μM for both SUMO1 and SUMO2.

In total, 10 µg of protein from 4-week-old adrenals were used for measurement with PKA kinase activity kit (ab139435, Abcam).

## In vitro SUMOylation assay

Protein purification (SUMO2, SAE1/UBA2, UBC9) and in vitro SUMOylation reactions were performed as previously described[73]. Recombinant PRKACA (100 ng) or PRKACB (100 ng) were incubated with 150 ng SAE1/UBA2, 225 ng UBC9, 1 µg SUMO2 in a total volume of 15 µL of 20 mM HEPES, pH7.3, 110 mM KOAc, 2 mM Mg(OAc)$_2$ 0.5 mM EGTA, 0.05% Tween-20, 0.2 mg/mL ovalbumin, 1 mM DTT and 1 mg/mL each of leupeptin, aprotinin and pepstatin). ATP (100 µM) was added to start the reaction, which was incubated at 37 °C for 1 h. 50 nM final of recombinant SENP2 (catalytic domain, Boston Biochem) was added for 30 min at 37 °C. The reactions were stopped by adding 5 µL of Laemmli buffer 4×, loaded on SDS-PAGE and immunoblotted with PKA antibodies.

## Statistics and reproducibility

Statistics were conducted using R language[70] and Comp3Moy function from sumo package (https://github.com/Damien-Dufour/sumo). Normality of populations distribution was assessed with Shapiro & Wilk test for n∈[7,5000] or otherwise Kolmogorov–Smirnov normality test.

If data followed a normal distribution, homoscedasticity was estimated with a Barlett test. To compare two populations, unpaired, two-tailed $t$ test was used for normally distributed data with the same variance, Mann–Whitney for non-normal distributions and Welch $t$ test for normally distributed data but with different variances. To compare three or more distributions: one-way ANOVA for normally distributed samples with pairwise multiple $t$ tests or Kruskal–Wallis for non-normally distributed samples with planned comparisons using Dunn's test to determine the genotype effect or the treatment effect.

Crosses on the violin plots represent the mean and lines represent the median. Error bars in barplot represent the SD unless otherwise stated. The number of samples per condition is indicated at the bottom of each plot. Corticosterone response to ACTH in Fig. 5I was analysed with paired two-way ANOVA to compare the effect of treatment for each genotype. Correlations in Figs. 6f and 7a have been conducted using cor.test function from stats package and the Kendall method as distributions did not follow normality and contained ties. All experiments have been repeated at least twice of thrice with consistent results. Immunostaining pictures are representative of a group of at least five replicates.

## Reporting summary

Further information on research design is available in the Nature Portfolio Reporting Summary linked to this article.

## Data availability

Sequencing data have been deposited in GEO with the accession code GSE193480. All other data that support the findings of this study are provided in the article or supplementary data. Source data are provided with this paper.

## Code availability

The code used to generate Figs. 4c, 7a and Supplementary Figs. 7b, 8c and 8d has been deposited on GitHub and is available through Zenodo[74] and the hyperlink https://zenodo.org/record/7347553. All other codes for data cleaning and analysis associated with the current submission are available upon request without restrictions.

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

## Acknowledgements
We thank K. Ouchen, S. Plantade, and P. Mazuel for animal care, A. Dehaze and J.P. Saru for their technical assistance, and Y. Renaud for the management of the bioinformatic platform. This work was funded through institutional support from CNRS, INSERM, Université Clermont-Auvergne, the French government IDEX-ISITE initiative 16-IDEX-0001 (CAP 20-25), and grants from Ministère de l'Enseignement Supérieur, de la Recherche et de l'Innovation (to D.D.), Société Française d'Endocrinologie (to D.D. and A.M.), Fondation Association pour la Recherche sur le Cancer (to J.W. and J.O.), and Agence Nationale pour la Recherche (ANR-18-CE14-0012-02 Sex-Specs to A.L. and A.M.).

## Author contributions
D.D., T.D., G.B. and A.M. designed research. D.T.B. and E.T.H. provided *AS^{Cre}*-expressing and *Senp2*-floxed mice, respectively. D.D., I.S.B., A.C., M.O., E.P., J.J.W., J.O., C.L., A.L., C.S.D., J.C.P. and F.R.B. performed experiments. D.D. and A.M. analysed the data. D.D. and A.M. wrote the paper. G.B., I.T., D.T.B., P.V. and A.M.L.M. edited the paper.

## Competing interests
The authors declare no competing interests.

## Additional information

[1]institut Génétique, Reproduction & Développement (iGReD), CNRS, INSERM, Université Clermont Auvergne, Clermont–Ferrand F-63000, France. [2]Department of Internal Medicine, Division of Metabolism, Endocrinology, and Diabetes, University of Michigan, Ann Arbor, MI, USA. [3]Training Program in Organogenesis, Center for Cell Plasticity and Organ Design, University of Michigan, Ann Arbor, MI, USA. [4]IGMM, Université de Montpellier, CNRS, Montpellier, France. [5]Service de Génétique Moléculaire, Pharmacogénétique et Hormonologie, Hôpital de Bicêtre, Assistance Publique-Hôpitaux de Paris (APHP), Physiologie et Physiopathologie Endocriniennes, INSERM, Université Paris-Saclay, Le Kremlin-Bicêtre, France. [6]Endocrinologie Moléculaire et Maladies Rares, Centre Hospitalier Universitaire, Université Claude Bernard Lyon 1, Bron, France. [7]Service d'Endocrinologie, Centre Hospitalier Universitaire Gabriel Montpied, Université Clermont Auvergne, Clermont-Ferrand, France. [8]Department of Internal Medicine, University of Arkansas for Medical Sciences, Little Rock, AR, USA. [9]Division of Endocrinology, Boston Children's Hospital, Department of Pediatrics, Harvard Medical School, Boston, MA, USA. [10]Harvard Stem Cell Institute, Harvard University, Cambridge, MA, USA. ✉e-mail: antoine.martinez@uca.fr

