## [Peer Review File · Nature Communications]

Loss of SUMO-specific protease 2 causes isolated glucocorticoid deficiency by blocking adrenal cortex zonal transdifferentiation in miceREVIEWER COMMENTS

Reviewer #1 (Remarks to the Author):

The manuscript by Dufour D et al “Loss of SUMO-specific protease 2 causes isolated glu- cocorticoid deficiency by blocking adrenal cortex zonal transdifferentiation” investigates the role of SUMOylation in adrenocortical function via conditional deletion of the deSUMOylating enzyme SUMO-specific protease 2 (SEN2) in steroidogenic cells (SEN2cKO). The group previously reported a gradient of protein SUMOylation in human and mouse normal adrenals, the highest being in the Zona Glomerulosa (ZG) and the lowest in the inner Zona Fasciculata (ZF), gradient that was found to be modulated by PKA or WNT signalling pathways, physiologically and in adrenal disorders.

Here, ZF-specific hypoplasia was observed in SEN2cKO mice, developing between 4 and 8 weeks in both males and females, while ZG morphology (and function) was largely unaffected. Interestingly, in their experimental set-up, clusters of dysplastic/hyperplastic cells were observed in the ZF, mostly in females, clusters composed of cells which had seemingly escaped recombination, as demonstrated by lineage tracing. Moreover, these hyperplastic clusters expressed nuclear SUMO2/3 and higher levels of a ZF marker, while recombined areas (hypoplastic) showed faint and cytoplasmic SUMO2/3 expression, and lower levels of ZF markers.

ZF atrophy was accompanied by corticosterone deficiency at 4 weeks in males, while aldosterone levels did not change between WT and SEN2cKO mice. In males, plasma corticosterone levelled between WT and SEN2cKO at later stages, suggesting a role of hyperplastic clusters in counterbalancing corticosterone loss in the hypoplastic areas of the ZF. In both sexes, high circulating ACTH levels were observed, showing isolated GC deficiency/ACTH insensitivity, and further supported by a blunted synacthen test response in SEN2cKO mice. The involvement of PKA activity was assessed by employing mice without (the PKA) R1 α subunit, resulting in higher PKA activity, in a SEN2cKO background. PKA de-repression was unable to restore the ZF phenotype, and actually absence of SEN2 lead to repression of PKA itself, strongly suggesting a direct effect of SUMOylation in the regulation of PKA activity in the ZF. Further, an improper centripetal migration/differentiation was observed in SEN2cKO mice using the AS Cre X SEN2cKO R26RmTmG/mTmG strain, with lower cells with ZF identity. Very interestingly, apoptotic cells were found to be in a very unusual area in the SEN2cKO adrenal, namely at the ZG/ZF boundary (as opposed to the inner ZF/medulla boundary in WT mice), and mechanistically this was linked to a blunted Ser637 phosphorylation of DRP1. Finally, by employing comparative bulk RNASeq analysis between WT and SEN2cKO adrenals the authors confirmed downregulation of steroid hormone biosynthesis pathway in SEN2cKO adrenals, whilst observing an enrichment of WNT signaling. They also observed ectopic accumulation of nuclear beta catenin in cells with ZF identity in SEN2cKO adrenals, and, and an enhanced SUMO2/3ylation of beta catenin.

This paper is well written, includes a solid dataset using appropriate mouse models, in vitro models, power, and statistical analysis. The authors do an excellent job in demonstrating the profound effect that SEN2 deletion has on steroidogenic function in the zona fasciculata.

Even if the obvious and immediate beneficiaries of this research are basic and translational biologists

studying the adrenal cortex, their finding on the role of SUMOylation in modulating the balance of PKA/WNT pathways, transdifferentiation processes and ultimately, tissue homeostasis has far reaching impacts for the wider scientific community, especially those with an interest in self-renewing organs and lineage commitment.

I have the following suggestions:

1) Fascinating is the observation of cells escaping recombination to counteract GC deficiency and would be interesting to further assess differences between these clusters and neighbouring hypoplastic areas in terms of lipid contents. For example, do GFPneg clusters become lipid enriched compared to SENP2-KO ZF hypoplastic cells? I would propose an Oil Red O staining or electron microscopy.

2) I found the data on adrenocortical progenitor cell expression interesting (Fig S4, RT-qPCR on both capsular, such as Gli1 and Dlk1) and subcapsular (Shh, Wnt4) markers. This is backed up by their lineage tracing using the B2/AS Cre. Even if the analysis was collected at 4 weeks, I presume a good proportion of mice at 40 weeks (although not aged yet) have developed subcapsular hyperplasia in their adrenals. The authors previously provided convincing data of the steroidogenic origin of these cells (ex SF-1 positive cells dedifferentiating). Did the authors observe any difference in the number, size, phenotype of these fibroblasts-like clusters in SENP2cKO adrenals? The observed higher expression of progenitor cell markers could also be due to their accumulation in these structures.

Reviewer #2 (Remarks to the Author):

Dufour et al are reporting in this manuscript that SUMO-specific protease 2 (SENP2) is crucial in murine adrenal cortex development. For this they developed mouse models of adrenal hyperSUMOylation by conditional ablation of Senp2 in adrenal cortex cells (actually in SF-1 expressing cells). They conclude that SENP2 loss shifts the balance between ACTH/PKA and WNT/beta-catenin signalling leading to a block in trans-differentiation between Glomerulosa to Fasciculata zones and thus glucocorticoid deficiency.

The study is very interesting, the paper is well written and the authors screened extensively for factors and processes that might be involved.

However there are a few aspects that were omitted from the paper that may improve an already impressive study.

1. A minor point actually but quite important, in the introduction, but also throughout the text, the Authors speak generally about the adrenal cortex development model. This may lead the reader to think that this refers to the human adrenal cortex. However, the mechanism presented and most of the data cited are referring to the murine adrenal cortex development (obvious also by the fact that the model

mentions only two adrenal cortex zones as compared to the three zones in humans). This should be specified as this model has not been, at least not yet, shown to be completely transferable to the human adrenal cortex development.

2. The conditional *Senp2* ablation mice were created by crossbreeding floxed *Senp2* mice with SF-1/NR5A1-Cre mice. The authors say that this is a adrenal specific recombination. However, the *Senp2* ablation would take effect in all the cells expressing SF-1 and these are not restricted to the cells of the adrenal cortex but include also neurons in the ventromedial hypothalamic nucleus as well as pituitary and gonadal cells. The SF-1 cells in the pituitary will be the growth hormone (GH) secreting cells which play also an important role in tissue and organ development and the steroidogenic cells in the gonads are also important for gonadal development and sexual dimorphism. That is why it would be very interesting if the authors could also check the extent of *Senp2* ablation in these organs, whether there is an effect on the GH and/or sex hormones production. This may have an indirect effect on the adrenal cortex development.

3. The authors mention in the introduction that sumoylation plays an important role in SF-1 function. Therefore it is surprising that they did not look at the sumoylation and expression levels of this transcription factor that plays such a crucial role in adrenocortical development. Was it due to the genetic background of the mice (NR5A1-Cre)?

Reviewer #3 (Remarks to the Author):

The manuscript by Damien Dufour et al follow up their previous finding that SUMOylation in the human and mouse adrenal cortex is subject to hormonal and spatial control. First, they show that adrenal cortex-specific knockout in mice results in zF hypoplasia, glucocorticoid deficiency and impaired response to ACTH. SENP2 ablation also blocks zG to zF transdifferentiation. They show that SENP2 loss leads to decreased PKA kinase activity, thus reducing its targets' phosphorylation including CREB, TRIM28 and DRP1 and in turn compromising the expression of steroidogenic genes and increasing apoptosis at the zG-zF boundary. Furthermore, loss of SENP2 impacts on WNT/ β -catenin pathway, leading to accumulation of β -catenin in the zF and activation of WNT pathway.

Overall, this work describes interesting observations, but the data are correlative or indirect, lacking a focus on gaining further insight into the mechanism how SENP2, as a deSUMOylase, regulates the adrenal cortex zonation and the response to stress.

Major concerns:

- (1). Given that there are steroidogenic progenitor cells that escaped the Cre-mediated *Senp2* recombination and gave rise to bulk of WT *Senp2* zF in mutant adrenals, scRNA-seq or ATAC-Seq would be a better approach than bulk RNA seq to analyze the transcriptional profile or chromatin accessibility change induced by *Senp2* knockout.
- (2). The authors showed that SENP2 loss affects both the PKA kinase activity and β -catenin SUMOylation status. I would appreciate if they could show the physical interaction between SENP2 and PKA, and

SEN2 and β -catenin. Co-immunoprecipitation or PLA are all good approaches to prove the interaction. Furthermore, it would be interesting to test whether SEN2 can directly deSUMOylate both PKA and β -catenin by in cell or in vitro assays.

(3). SEN2 has a broad deSUMOylation activity, it would be critical to address the question whether knockin of the SEN2 catalytic mutant C548S can cause the similar defects in adrenal cortex. This will reveal whether SEN2 enzymatic activity is playing a role in adrenal cortex zonation and stress response.

(4). I appreciate the effort of the authors to go into great detail about the balance between ACTH/PKA and WNT/ β -catenin signaling upon SEN2 loss. But it would be informative if authors could identify proteins whose SUMOylation can be affected or increased by SEN2 knockout in adrenal cortex using quantitative proteomics mass spec. This may help identify the direct protein targets that play roles in adrenal cortex zonal transdifferentiation.

(5). Throughout the paper, authors only showed SEN2 mRNA reduction by about 50% and genomic PCR to prove the Senp2 knockout. It would be great if authors also show the SEN2 protein reduction in adrenal cortex by both western blot and immunostaining.

Minor points:

(1). Fig 4 F showed the IF images of AKR1B7 and GFP from 24-week-old female mice, but the main text described the mice were 40-week-old.

(2). Fig 7C top and middle panels are from 4-week-old mice, whereas the bottom panels are taken from 24-week-old mice.

Point-by-point response to the reviewers' comments

We thank the reviewers for evaluating this work and for guiding us, through their constructive criticism, toward a revised version of the manuscript that is scientifically strengthened. We believe that we have addressed most of the reviewers' concerns by providing new experimental data that reinforce our conclusions and where this was not possible/feasible, we have provided carefully argued responses.

This led to the addition in the revised version of new figures (Fig. 2.E. & Fig. 4) and supplemental figures (S1.E.; S2; S7.C.; S8), and inclusion of new paragraphs in the Results section (p7, p10-11, p12-13, p16-17, p22), in the Discussion section (p28, p30) and in the Methods section (p32, p34, p37). To help track changes in the text, any modification in all files including supplementary information was written in red.

Reviewer #1 (remarks to the Authors)

The manuscript by Dufour D et al "Loss of SUMO-specific protease 2 causes isolated glucocorticoid deficiency by blocking adrenal cortex zonal transdifferentiation" investigates the role of SUMOylation in adrenocortical function via conditional deletion of the deSUMOylating enzyme SUMO-specific protease 2 (SEN2) in steroidogenic cells (SEN2cKO). The group previously reported a gradient of protein SUMOylation in human and mouse normal adrenals, the highest being in the Zona Glomerulosa (ZG) and the lowest in the inner Zona Fasciculata (ZF), gradient that was found to be modulated by PKA or WNT signalling pathways, physiologically and in adrenal disorders.

Here, ZF-specific hypoplasia was observed in SEN2cKO mice, developing between 4 and 8 weeks in both males and females, while ZG morphology (and function) was largely unaffected. Interestingly, in their experimental set-up, clusters of dysplastic/hyperplastic cells were observed in the ZF, mostly in females, clusters composed of cells which had seemingly escaped recombination, as demonstrated by lineage tracing. Moreover, these hyperplastic clusters expressed nuclear SUMO2/3 and higher levels of a ZF marker, while recombined areas (hypoplastic) showed faint and cytoplasmic SUMO2/3 expression, and lower levels of ZF markers.

ZF atrophy was accompanied by corticosterone deficiency at 4 weeks in males, while aldosterone levels did not change between WT and SEN2cKO mice. In males, plasma corticosterone levelled between WT and SEN2cKO at later stages, suggesting a role of hyperplastic clusters in counterbalancing corticosterone loss in the hypoplastic areas of the ZF. In both sexes, high circulating ACTH levels were observed, showing isolated GC deficiency/ACTH insensitivity, and further supported by a blunted synacthen test response in SEN2cKO mice. The involvement of PKA activity was assessed by employing mice without (the PKA) R1 α subunit, resulting in higher PKA activity, in a SEN2cKO background. PKA de-repression was unable to restore the ZF phenotype, and actually absence of SEN2 lead to

repression of PKA itself, strongly suggesting a direct effect of SUMOylation in the regulation of PKA activity in the ZF. Further, an improper centripetal migration/differentiation was observed in SENP2cKO

mice using the AS Cre X SENP2cKO R26RmTmG/mTmG strain, with lower cells with ZF identity. Very interestingly, apoptotic cells were found to be in a very unusual area in the SENP2cKO adrenal, namely at the ZG/ZF boundary (as opposed to the inner ZF/medulla boundary in WT mice), and mechanistically this was linked to a blunted Ser637 phosphorylation of DRP1. Finally, by employing comparative bulk RNASeq analysis between WT and SENP2cKO adrenals the authors confirmed downregulation of steroid hormone biosynthesis pathway in SENP2cKO adrenals, whilst observing an enrichment of WNT signaling. They also observed ectopic accumulation of nuclear beta catenin in cells with ZF identity in SENP2cKO adrenals, and, and an enhanced SUMO2/3ylation of beta catenin.

This paper is well written, includes a solid dataset using appropriate mouse models, in vitro models, power, and statistical analysis. The authors do an excellent job in demonstrating the profound effect that SENP2 deletion has on steroidogenic function in the zona fasciculata.

Even if the obvious and immediate beneficiaries of this research are basic and translational biologists studying the adrenal cortex, their finding on the role of SUMOylation in modulating the balance of PKA/WNT pathways, transdifferentiation processes and ultimately, tissue homeostasis has far reaching impacts for the wider scientific community, especially those with an interest in self-renewing organs and lineage commitment.

I have the following suggestions:

1) Fascinating is the observation of cells escaping recombination to counteract GC deficiency and would be interesting to further assess differences between these clusters and neighbouring hypoplastic areas in terms of lipid contents. For example, do GFPneg clusters become lipid enriched compared to SENP2-KO ZF hypoplastic cells? I would propose an Oil Red O staining or electron microscopy.

Response - Revised Figure 2E : We agree with the reviewer's comment and we have now provided additional analyses of lipid contents on adrenal cryosections using Bodipy fluorescent staining (allowing neutral lipid staining just as ORO). These experiments have been included in the Revised Figure 2E and show that hyperplastic areas containing cells escaping recombination (GFP-) are enriched in neutral lipids when compared to neighboring hypoplastic GFP+ areas. This likely reflects the over-stimulation of these cells which, in limited numbers, must respond to the elevated levels of circulating ACTH to compensate for the loss of the zF and maintain glucocorticoid homeostasis.

Corresponding comments have been integrated in the result section at the bottom of p12 and top of p13.

2) I found the data on adrenocortical progenitor cell expression interesting (Fig S4, RT-qPCR on both capsular, such as Gli1 and Dlk1) and subcapsular (Shh, Wnt4) markers. This is

backed up by their lineage tracing using the B2/AS Cre. Even if the analysis was collected at 4 weeks, I presume a good proportion of mice at 40 weeks (although not aged yet) have developed subcapsular hyperplasia in their adrenals. The authors previously provided convincing data of the steroidogenic origin of these cells (ex SF-1positive cells dedifferentiating). Did the authors observe any difference in the number, size, phenotype of these fibroblasts-like clusters in SENP2cKO adrenals? The observed higher expression of progenitor cell markers could also be due to their accumulation in these structures.

Response - Figure to reviewer #1: the reviewer refers to the spindle-shaped cells spilling into the subcapsular area with age. As requested, we have quantified in 40-week-old mice, on the one hand, the prevalence of adrenals with spindle-shaped cells and on the other hand, the surface area of these cell clusters as a proxy of cell number. We found that these fibroblasts-like clusters were more frequent and larger in WT females than in males but this gender dimorphism disappeared in *Senp2^{ckO}* due to decreased prevalence and size in females. Thus, the observed increased in capsule thickness and progenitor markers expression obviously did not result from the accumulation of spindle-shaped cell structures.

Reviewer #2 (remarks to the Authors)

Dufour et al are reporting in this manuscript that SUMO-specific protease 2 (SEN2) is crucial in murine adrenal cortex development. For this they developed mouse models of adrenal hyperSUMOylation by conditional ablation of *Senp2* in adrenal cortex cells (actually in SF-1 expressing cells). They conclude that SEN2 loss shifts the balance between ACTH/PKA and WNT/beta-catenin signalling leading to a block in trans-differentiation between Glomerulosa to Fasciculata zones and thus glucocorticoid deficiency.

The study is very interesting, the paper is well written and the authors screened extensively for factors and processes that might be involved.

However there are a few aspects that were omitted from the paper that may improve an already impressive study.

1. A minor point actually but quite important, in the introduction, but also throughout the text, the Authors speak generally about the adrenal cortex development model. This may lead the reader to think that this refers to the human adrenal cortex. However, the mechanism

presented and most of the data cited are referring to the murine adrenal cortex development (obvious also by the fact that the model mentions only two adrenal cortex zones as compared to the three zones in humans). This should be specified as this model has not been, at least not yet, shown to be completely transferable to the human adrenal cortex development.

Response - To remove any ambiguity concerning the mechanisms of adrenal cortex development to which we refer, which were essentially demonstrated in mice, we have added the words "murine" or "mouse" in the introduction section p4 (first 2 paragraphs) and in the discussion p30 (last paragraph).

2. The conditional *Senp2* ablation mice were created by crossbreeding floxed *Senp2* mice with SF-1/NR5A1-Cre mice. The authors say that this is a adrenal specific recombination. However, the *Senp2* ablation would take effect in all the cells expressing SF-1 and these are not restricted to the cells of the adrenal cortex but include also neurons in the ventromedial hypothalamic nucleus as well as pituitary and gonadal cells. The SF-1 cells in the pituitary will be the growth hormone (GH) secreting cells which play also an important role in tissue and organ development and the steroidogenic cells in the gonads are also important for gonadal development and sexual dimorphism. That is why it would be very interesting if the authors could also check the extent of *Senp2* ablation in these organs, whether there is an effect on the GH and/or sex hormones production. This may have an indirect effect on the adrenal cortex development.

Response – Revised Supplemental figure S2: SF-1 is not expressed in somatotrophs, but in gonadotrophs (see Ref 23 & 24). In any case, we agree that it is important to exclude any other organ influence where Cre-mediated recombination might take place to better interpret the adrenal phenotype. So, we performed IHF/IHC analyses of pituitaries that show no changes in gonadotrophs number or organization (Foxl2+) in *Senp2^{ckO}* compared to WT. Testosterone plasma levels in males and progesterone in females were unaffected, indicating no major change in gonadal steroids production in *Senp2^{ckO}* mice. However, reproductive success was altered in mutant females, and gonadal weights were decreased in both genders. Impacts of *Senp2* ablation on gonads histology is illustrated by H&E sections from mutant testis and ovary. In line with these data, FSH plasma levels were low in mutant males whereas LH levels were increased in females, confirming that gonadal activity was affected by *Senp2* ablation. The gonadal phenotype of *Senp2^{ckO}* mice is currently the subject of intensive studies by our team and is outside the scope of the present paper. In conclusion, gonadal phenotype is unlikely to explain adrenal insufficiency/atrophy in *Senp2^{ckO}* mice although we did not exclude an indirect effect of sex steroids to explain the better ability of female adrenals to escape *Senp2* ablation (comments of Figure 1F and conclusion of the 1st paragraph of results). Finally, we also provided demonstration that block of zF transdifferentiation from zG and deficit in ACTH responsiveness also occurred in mice with postnatal adrenal-specific *Senp2* invalidation (*AS-Senp2^{ckO}*) (Figure 5E-H).

Comments corresponding to supplemental Figure S2 have been introduced in p10-11 of the revised manuscript.

3. The authors mention in the introduction that sumoylation plays an important role in SF-1 function. Therefore it is surprising that they did not look at the sumoylation and expression levels of this transcription factor that plays such a crucial role in adrenocortical development. Was it due to the genetic background of the mice (NR5A1-Cre)?

Response - Supplemental figure S8: we agree with the reviewer's comment and as initially illustrated in supplemental figure S5.B., SF-1 mRNA expression levels were slightly upregulated in males and unchanged in females upon *Senp2* invalidation. Similarly, SF-1 immunostaining didn't change when comparing WT and mutant adrenals (Revised supplemental figure S8.B.). So, adrenal insufficiency found in mutants is unlikely to rely on decreased SF-1 expression levels.

However, it is not excluded that some changes in the expression of downstream SF-1 targets could reflect the expected increase in SUMOylation of SF-1. However, among the very few identified SUMO-sensitive genes, whose expression was enhanced by unSUMOylatable SF-1 (SF-1^{2KR}), including *Inha*, *Shh* and *Sox9* (Campbel et al 2008 PMID:18838537 and ref 21), none of them were down-regulated as would be expected if SF-1 was more SUMOylated in *Senp2*^{CKO} adrenals. Indeed, *Inha* mRNA levels were unaffected while both *Shh* (and its target *Gli1*) and *Sox9* were upregulated (revised supplemental figure S8.A.). So, this would rather suggest that SF-1 behaved as being hypoSUMOylated in *Senp2*^{CKO} adrenals. In addition, we used PLA analyses to quantify SF-1 SUMOylation *in situ* and found very few SUMOylated SF-1 foci which number was not affected by SENP2 loss (Supplemental figure S8.C.D.).

These data and corresponding comments have been included as part of the discussion in the middle of p30.

Reviewer #3 (Remarks to the Author):

The manuscript by Damien Dufour et al follow up their previous finding that SUMOylation in the human and mouse adrenal cortex is subject to hormonal and spatial control. First, they show that adrenal cortex-specific knockout in mice results in zF hypoplasia, glucocorticoid deficiency and impaired response to ACTH. SENP2 ablation also blocks zG to zF transdifferentiation. They show that SENP2 loss leads to decreased PKA kinase activity, thus reducing its targets' phosphorylation including CREB, TRIM28 and DRP1 and in turn compromising the expression of steroidogenic genes and increasing apoptosis at the zG-zF boundary. Furthermore, loss of SENP2 impacts on WNT/ β -catenin pathway, leading to accumulation of β -catenin in the zF and activation of WNT pathway.

Overall, this work describes interesting observations, but the data are correlative or indirect, lacking a focus on gaining further insight into the mechanism how SENP2, as a deSUMOylase, regulates the adrenal cortex zonation and the response to stress.

Major concerns:

(1). Given that there are steroidogenic progenitor cells that escaped the Cre-mediated *Senp2* recombination and gave rise to bulk of WT *Senp2* zF in mutant adrenals, scRNA-seq or ATACSeq would be a better approach than bulk RNA seq to analyze the transcriptional profile or chromatin accessibility change induced by *Senp2* knockout.

Response – The reviewer’s concern is important, but the requested single cell (sc) analyses are really premature at this stage of the phenotypic study of our animal model. Furthermore, we would like to bring to the reviewer's attention that the analysis of chromatin accessibility is not the object of this study and even if the transcriptomic analysis allowed to confirm the alteration of WNT signaling, it is not the starting point of our phenotypic analysis which relies mainly on a deductive reasoning based on candidate mechanisms, well recognized in adrenal pathophysiology.

Although some cells escape Cre-mediated recombination (as it has been carefully examined in our manuscript), our global adrenal transcriptome analysis performed at 4 weeks of age is fully justified here. Indeed, as illustrated by Supplemental figure S7 including principal component analysis, the transcriptomes of WT and mutant adrenals are grouped in distinct clusters, indicating that our global data are valid for tracking the transcriptional consequences of *Senp2* invalidation.

In absolute terms, the sc approach is the best way to identify cell specific molecular signatures linked to gene ablation studies. Even leaving aside the fact that the enzymatic cell dissociation of adult adrenal glands poses yet unsolved technical challenges (important biases in the representation of the different cell types), the logical extension of our studies will be to identify changes in chromatin accessibility and chromatin SUMOylation landscape in whole *Senp2^{ckO}* adrenals. However, we think that sc experiments whether they are applied to the transcriptome and a fortiori to chromatin analysis, or SUMO cistrome are out of the scope of the present paper and above all, technically unfeasible at the moment.

(2). The authors showed that SENP2 loss affects both the PKA kinase activity and β -catenin SUMOylation status. I would appreciate if they could show the physical interaction between SENP2 and PKA, and SENP2 and β -catenin. Co-immunoprecipitation or PLA are all good approaches to prove the interaction. Furthermore, it would be interesting to test whether SENP2 can directly deSUMOylate both PKA and β -catenin by in cell or in vitro assays.

Response - revised Figure 4: we agree that showing SUMO modification of PKA and regulation by SENP2 is an important point to address. We have now performed in vitro SUMOylation assays showing SUMO conjugation of Ca and Cb catalytic subunits of PKA and their deconjugation by SENP2. Furthermore, we confirmed *in vivo* using PLA, that loss of *Senp2* doubles the number of cortical cells bearing SUMOylated Cab catalytic subunits. Finally, we used the selective SUMO E1 inhibitor TAK-981 to show that pharmacological attenuation of SUMOylation *in vivo* in mice, improves ACTH/PKA responsiveness in WT and mutants. Overall, these converging experiments illustrated in revised figure 4 demonstrate that loss of SENP2 represses PKA activity by increasing its SUMOylation. Corresponding comments have been introduced on p16-17.

In contrast, we estimate that proofs provided in this manuscript (figure 8) for in vivo SUMOylation of b-catenin and increased conjugation, following SENP2 loss are sufficient (IHC, reciprocal coimmunoprecipitations and PLA, all conducted *in vivo*). Whether SENP2 directly deconjugates or indirectly alters b-catenin SUMOylation is not an essential question to address here.

(3). SENP2 has a broad deSUMOylation activity, it would be critical to address the question whether knockin of the SENP2 catalytic mutant C548S can cause the similar defects in adrenal cortex. This will reveal whether SENP2 enzymatic activity is playing a role in adrenal cortex zonation and stress response.

Response - To our knowledge, C548S knock-in mouse model is not available, yet... the generation of such a model would require not only the introduction of suitable point mutations in the *Senp2* locus, but also conditioning the expression of this mutation on the presence of the Cre recombinase, since whole body homozygous mutation is expected to be detrimental for embryonic development. This will take years and is really not reasonable. As a matter of fact, all available models for *Senp2* conditional alleles were designed to inactivate SUMO protease domain encoded by exons 12-16 through floxing either exons 13-14 (Yeh's lab model, this paper) or exon 16 (Hsu's lab model). In line with this, mutant *Senp2* transcripts still accumulate in *Senp2*^{CKO} adrenals and therefore exons 13-14 deletion does not lead to mRNA decay (see revised supplemental Figure S7.C and corresponding comments on p22 where RNAseq coverage plots show as expected, a decreased in the number of reads in the sequences of exons 13 and 14 and none in the upstream or downstream exons).

Furthermore, although we understand that SENP2 may have functions outside its protease domain, the specific deletion of exon 16 (*Senp2*^{SUMO} model) was shown to phenocopy extraembryonic and embryonic defects of *Senp2*-null allele (Fu et al. PLOS Genet. 2014; PMID: 25299344).

Finally, we provided new in vivo experiments using a selective SUMO E1 inhibitor that together with other converging data strongly argues for a central role of SENP2 enzymatic activity in the regulation of adrenal stress response (revised Figure 4.D-G and corresponding comments on p17 and p28).

(4). I appreciate the effort of the authors to go into great detail about the balance between ACTH/PKA and WNT/ β -catenin signaling upon SENP2 loss. But it would be informative if authors could identify proteins whose SUMOylation can be affected or increased by SENP2 knockout in adrenal cortex using quantitative proteomics mass spec. This may help identify the direct protein targets that play roles in adrenal cortex zonal transdifferentiation.

Response – Experimental evidence from different groups including ours, showed that PKA and WNT signalling pathways are key players of adrenal zonation, homeostasis, function and pathology (ref 3-9). It was thus essential to show how they can be regulated by SUMOylation.

This is what we did in this paper, by identifying how invalidation of SENP2 functionally alters these signalling pathways.

We agree that an unbiased analysis of the adrenal SUMO proteome would be very informative to get a broader view of the mechanisms involved in zonal transdifferentiation defects. This is clearly another project outside the scope of this article, which has established the proof of concept of the *in vivo* importance of SUMOylation in adrenal function that is a prerequisite for identifying the conditions for future SUMOylome studies.

Although we have planned quantitative proteomic analyses from mutant and WT adrenals (under basal or stress conditions), specific conditions must first be met to adapt these analyses to such a small gland (1-2mg per gland) and facilitate the enrichment of SUMOylated peptides. This will be achieved by using the CAG-SUMO transgenic mice from Yang *et al.* 2014 (PMID: 24569813) available at Jackson's lab and will be the subject of a future project.

(5). Throughout the paper, authors only showed SENP2 mRNA reduction by about 50% and genomic PCR to prove the *Senp2* knockout. It would be great if authors also show the SENP2 protein reduction in adrenal cortex by both western blot and immunostaining.

Response – Revised supplemental figure S1.E: We have now provided information showing decreased accumulation of adrenal SENP2 protein by WB. Corresponding comments introduced on p7.

Minor points:

(1). Fig 4 F showed the IF images of AKR1B7 and GFP from 24-week-old female mice, but the main text described the mice were 40-week-old.

Response - We have replaced 40 by 24 in the revised text (p19) in order to match the IF image of Figure 4.F. (now renumbered Figure 5.F.)

(2). Fig 7C top and middle panels are from 4-week-old mice, whereas the bottom panels are taken from 24-week-old mice.

Response - IF illustration of bottom panel has been replaced by using adrenal section from 4week-old mice in revised Figure 7.C. (now renumbered Figure 8.C.).

REVIEWERS' COMMENTS

Reviewer #1 (Remarks to the Author):

RE: revision of revised manuscript by Dufour D et al “Loss of SUMO-specific protease 2 causes isolated glucocorticoid deficiency by blocking adrenal cortex zonal transdifferentiation”.

The authors have addressed the suggestions outlined after the first submission months ago; this has been done by either performing new experiments and by further analysis of their existing data/slides. This reviewer appreciate the effort by the authors. This manuscript is of high quality and worthy publishing in Nat Comm.

I have, obviously, attentively scrutinised the comments and criticisms made by the other reviewers and found the authors'reply very satisfactory.

Reviewer #2 (Remarks to the Author):

The authors addressed all my comments in a satisfactory manner. I want to apologize for my mistake with the SF-1 expression in somatotroph instead of gonadotroph cells, I realized my mistake in the second I sent the report however it seems that a correction was not possible as I received no feedback from the editors on my inquiry aimed to change my comment. However the authors understood and addressed this point very professionally and thoroughly despite my mistake.

Reviewer #3 (Remarks to the Author):

The authors have addressed most of my comments and provided appropriate new data or explanation. The newly added data fully support their conclusions and have made the manuscript much more robust.

Point-by-point response to the reviewers' comments

We thank the reviewers for evaluating this work and for guiding us, through their constructive criticism, toward a revised version of the manuscript that is scientifically strengthened.

REVIEWERS' COMMENTS

Reviewer #1 (Remarks to the Author):

RE: revision of revised manuscript by Dufour D et al "Loss of SUMO-specific protease 2 causes isolated glucocorticoid deficiency by blocking adrenal cortex zonal transdifferentiation".

The authors have addressed the suggestions outlined after the first submission months ago; this has been done by either performing new experiments and by further analysis of their existing

data/slides. This reviewer appreciate the effort by the authors. This manuscript is of high quality and worthy publishing in Nat Comm.

I have, obviously, attentively scrutinised the comments and criticisms made by the other reviewers and found the authors'reply very satisfactory.

Reviewer #2 (Remarks to the Author):

The authors addressed all my comments in a satisfactory manner. I want to apologize for my mistake with the SF-1 expression in somatotroph instead of gonadotroph cells, I realized my mistake in

the second I sent the report however it seems that a correction was not possible as I received no feedback from the editors on my inquiry aimed to change my comment. However the authors

understood and addressed this point very professionally and thoroughly despite my mistake.

Reviewer #3 (Remarks to the Author):

The authors have addressed most of my comments and provided appropriate new data or explanation. The newly added data fully support their conclusions and have made the manuscript much more robust.

Response – We thank all of the reviewers for their final positive evaluation of our revised manuscript and are truly grateful to each of them for the time they devoted to the detailed analysis of this work. As always, this is a painful moment for authors but one that, fortunately and invariably, leads to more good science.